# Formally Exploring Visual Anomaly Detection Evaluation Metrics

**Nasar Iqbal** [*1]   **Dennis Wagner** [*2]   **Philipp Liznerski** [2]   **Nabeel Hussain Syed** [2]   **Sophie Fellenz** [2]   **Niki Martinel** [1]
**Marius Kloft** [2]

## Abstract

Inaccurate Visual Anomaly Detection (VAD) can lead to critical failures in safety-sensitive domains, including autonomous navigation and industrial surveillance. Despite the abundance of VAD algorithms, assessing their capabilities remains challenging, because standard evaluation metrics often yield inconsistent or misleading results. In this paper, we demonstrate that standard evaluation paradigms fail to adequately capture model performance, often overlooking critical errors such as redundant detections and false positive distributions. To address this, we formalize the requirements for VAD evaluation by introducing a set of verifiable properties of evaluation methods. With a systematic analysis of current state-of-the-art evaluation methods, we prove that none satisfy the full suite of proposed properties, revealing significant, inherent inconsistencies. To bridge this gap, we introduce SAAM-ALARM, a novel evaluation metric mathematically designed to satisfy these formal properties. Our results show that SAAM-ALARM provides a more nuanced and theoretically sound assessment, offering a superior standard for performance benchmarking in anomaly detection.

## 1. Introduction

VAD is a fundamental problem in computer vision with direct implications for quality control, safety, and security-critical applications (Huang et al., 2022). The task aims to identify regions or patterns within an image that deviate from the expected appearance of normal data, such as cracks, scratches, cuts, or structural defects (Zipfel et al.,

---

[*]Equal contribution   [1]Department of Mathematics, Computer Science and Physics, University of Udine, Udine, Italy [2]Department of Machine Learning, University of Kaiserslautern-Landau, Kaiserslautern, Germany . Correspondence to: Nasar Iqbal <iqbal.nasar@spes.uniud.it>.

*Proceedings of the $43^{rd}$ International Conference on Machine Learning*, Seoul, South Korea. PMLR 306, 2026. Copyright 2026 by the author(s).

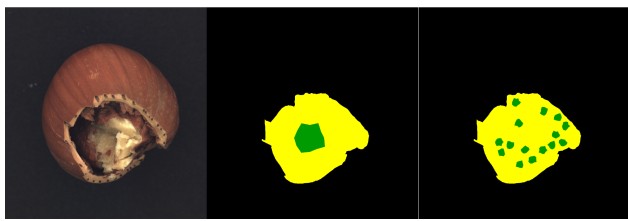

*Figure 1.* Motivating example illustrating redundant predictions. **Left**: Input image from the MVTec AD dataset. **Middle**: Model A produces a single coherent true positive. **Right**: Model B produces multiple fragmented true positives covering the same anomalous region. Despite identical or higher scores under standard metrics such as precision and recall, the fragmented prediction is harder to interpret and incurs unnecessary redundancy.

2023). Reliable detection of such anomalies is essential for developing trustworthy automated inspection systems that reduce human intervention while minimizing costly errors.

A central limitation of existing VAD metrics is their inability to account for redundancy in predicted anomaly regions. For example, two models may correctly localize the same defect, yet differ substantially in how it is represented: one produces a single coherent detection, while another fragments the region into multiple disconnected predictions. Standard metrics often assign equal or higher scores to fragmented predictions, despite their reduced interpretability and increased inspection effort, as illustrated in Fig. 1.

More fundamentally, commonly used VAD metrics implicitly encode assumptions about prediction quality, assumptions that are rarely made explicit or systematically verified. As a result, different metrics may produce contradictory rankings of methods while failing to penalize undesirable behaviors such as redundant or spatially incoherent predictions.

Recent work in time-series anomaly detection (Wagner et al., 2025) has shown that such inconsistencies are structural and can be revealed through formal, property-based analysis of evaluation metrics. Despite the growing maturity of VAD methods, an analogous formal framework for evaluating image-based anomaly detection metrics is still missing. Current practice relies largely on empirical benchmarking and convention-driven metric selection, offering limited insight into metric behavior and limitations.

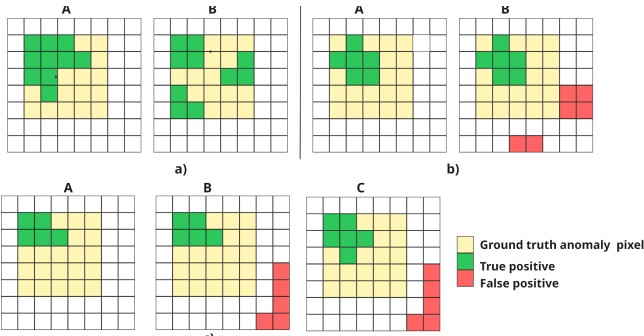

*Figure 2.* Structural failure modes of VAD metrics. **(a) Redundancy:** Conventional metrics fail to distinguish between cohesive and fragmented predictions, ignoring the industrial cost of redundant alarms. **(b) Recall Insensitivity:** Recall is agnostic to topological distribution, yielding identical scores for any spatial arrangement of a fixed True Positive count. **(c) Metric Saturation and Masking:** Illustrates how the mathematical structure of IoU and F1-score allows additional True Positive pixels to numerically compensate for False Positives. This "offset" effect results in identical scores for qualitatively different outputs, masking secondary false predictions that decrease operational reliability

In this work, we address this gap by introducing a formal, property-based framework for evaluating VAD metrics. We define a set of eight verifiable properties that capture essential spatial and structural requirements for visual anomaly evaluation. Using this framework, we conduct a rigorous assessment of six widely used threshold-dependent VAD metrics, demonstrating that none satisfy all proposed properties. Finally, we introduce a new evaluation metric that provably fulfills all properties, providing consistent and interpretable rankings aligned with the practical demands of visual inspection.

Our contributions are summarized as follows:

- *Formal framework:* We introduce a principled framework for evaluating VAD metrics by defining eight verifiable properties that formalize essential spatial and structural requirements.

- *Rigorous assessment:* We analyze six commonly used threshold-dependent VAD metrics and show that each violates one or more of the proposed properties.

- *New metric:* We propose a novel VAD metric that provably satisfies all defined properties, enabling more trustworthy and consistent evaluations.

## 2. Limitations of Existing Evaluation Metrics

Reliable evaluation is a fundamental requirement for VAD, as evaluation metrics directly shape how models are compared, selected, and deployed. While a wide range of quantitative metrics exists, each captures only specific aspects of prediction quality and may fail under common yet practically relevant prediction patterns, even when models correctly identify the same anomalous region. In this section, we analyze representative failure cases of commonly used pixel-level metrics to motivate the need for principled evaluation criteria.

In practice, VAD evaluation metrics are typically categorized as threshold-dependent or threshold-independent (Heckler-Kram et al., 2026). Threshold-dependent metrics, such as precision, recall, and F1-score (Wang et al., 2020), require selecting a decision threshold to classify samples or pixels as normal or anomalous. While these metrics are simple and computationally efficient, their reliability depends on the choice of threshold. In highly imbalanced industrial settings, where anomalous samples or pixels are extremely rare, small changes in threshold selection can lead to large and misleading variations in reported performance (Bergmann et al., 2021).

To mitigate sensitivity to threshold selection, threshold-independent metrics such as the Area Under the Receiver Operating Characteristic Curve (AUROC) (Diers & Pigorsch, 2023) and the Area Under the Precision–Recall Curve (AUPRC) are widely adopted in VAD benchmarks. While these aggregate performance across all thresholds, they serve as a high-level abstraction that can mask underlying structural issues in localization. In practical industrial settings, a decision threshold must eventually be applied to trigger an alarm or reject a part (Baitieva et al., 2025). Therefore, the reliability of the underlying threshold-dependent metrics remains the critical bottleneck for trustworthy inspection. In this work, we focus our formal analysis on these threshold-dependent measures to ensure they satisfy the spatial and structural requirements necessary for reliable defect localization.

We consider predictions produced under a shared experimental setting for industrial AD, such as MVTec-AD (Bergmann et al., 2019), and focus on widely adopted pixel-level metrics including precision, recall, Intersection over Union (IoU), and F1-score (Wang et al., 2020). These metrics are commonly used to evaluate localization performance in industrial inspection tasks, yet their behavior under structurally different but equally accurate predictions is often poorly understood. Our analysis adopts standard industrial inspection assumptions, including reliable ground-truth annotations and object-level alarm semantics, which we formalize in Section 4.

While the sensitivity of threshold-dependent metrics to class imbalance is well-documented (Wang et al., 2020), a more fundamental issue in VAD is their spatial insensitivity. Because these metrics treat pixels as independent units of evaluation, they remain blind to the topological structure of a prediction. For instance, the mathematical formulation of Recall, which is defined as the ratio of True Positive pixels to total Ground Truth pixels, is agnostic to whether those

pixels form a single coherent region or a fragmented set of disjointed points.

To isolate these structural effects, we compare predictions that cover the same ground-truth anomaly but differ in spatial organization. Under the assumption that each predicted connected component triggers an independent inspection action, these structural differences have significant practical consequences. Figure 2 illustrates several such prediction patterns. Precision penalizes predictions outside the anomalous region but remains unchanged when additional predictions fall entirely within the ground truth. Consequently, in Fig. 2(b), recall remains identical despite substantial differences in how the anomaly is represented, as recall depends only on the fraction of ground-truth anomaly pixels detected.

Metrics such as IoU and F1-score account for both false positives and false negatives by measuring overlap between predicted and ground-truth regions. However, Fig. 2(c) demonstrates that these metrics can be relatively insensitive to additional false predictions when overall overlap remains high. Moreover, small increases in true positive pixels, potentially due to incidental or random predictions, can offset false positives, yielding identical IoU or F1 scores for qualitatively different outputs.

Together, these examples highlight a fundamental limitation of existing pixel-level evaluation metrics:

**Limitation 1** (Lack of Structural Sensitivity). Under standard industrial inspection assumptions, commonly used pixel-level evaluation metrics fail to distinguish between structurally different anomaly predictions that localize the same defect, leading to identical or comparable scores for predictions with differing practical utility.

In real inspection settings, such insensitivity complicates result interpretation and downstream decision-making, as evaluation scores may fail to reflect differences that matter operationally. Together, these observations indicate that effective evaluation metrics for VAD must go beyond pixel-wise accuracy. In addition to identifying anomalous regions, metrics should explicitly account for structural properties of predictions under well-defined inspection assumptions. This motivates the need for explicitly stated and verifiable evaluation requirements, which we formalize in the following section.

## 3. Related Work

VAD has been studied under several closely related problem settings, which differ primarily in how anomalies are defined and what outputs are expected. In image-level AD, the task is to classify an entire image as normal or anomalous. In contrast, pixel-level or region-level AD requires producing a dense anomaly map that localizes defective regions, often framing the problem as a segmentation-like task. These distinctions directly influence evaluation protocols, as image-level detection emphasizes discrimination, while localization-oriented tasks require meaningful spatial assessment.

Industrial visual inspection has emerged as a dominant application of VAD, largely driven by benchmark datasets such as MVTec AD (Bergmann et al., 2019), which evaluates both image-level detection and pixel-level localization.

While this work focuses on evaluation rather than detection methods, it is worth noting that algorithmic developments have strongly influenced evaluation practice. A large body of work has focused on developing AD and localization algorithms, including reconstruction-based approaches such as autoencoders (Gong et al., 2019), and GAN-based models (Gu & Dao, 2024), teacher–student models (Tien et al., 2023), and feature-space nearest-neighbor techniques (Roth et al., 2022) leveraging pretrained backbones. Representative methods commonly used in benchmarks include PaDiM (Defard et al., 2021), PatchCore (Roth et al., 2022), and DRAEM (Zavrtanik et al., 2021). While these works primarily aim to improve detection performance, they have also implicitly shaped evaluation practice by standardizing reporting conventions (image-level versus pixel-level metrics) and popularizing specific evaluation measures, most notably AUROC-based scores.

**Classification-Based and ROC-Based Metrics** Early evaluation in VAD followed standard binary classification protocols, utilizing point-based metrics such as precision, recall, and the F1-score (Wang et al., 2020). As the field moved toward standardized benchmarking, the focus shifted toward threshold-independent measures to facilitate model comparison without committing to specific operating points.

Consequently, Receiver Operating Characteristic (ROC) curves and the Area Under the ROC Curve (AUROC) have become the de facto standard in the literature. Image-level AUROC (I-AUROC) assesses global discrimination, while pixel-level AUROC (P-AUROC) evaluates dense anomaly maps by aggregating pixel-wise scores across all possible thresholds. While these metrics are widely adopted for their ability to summarize performance, they fundamentally treat pixels as independent samples. This pixel-wise independence can mask significant qualitative differences in spatial coherence or fragmentation, which are often critical for successful industrial localization.

**Localization-Oriented Metrics** To better assess localization quality, several works adopt segmentation-style metrics such as Intersection over Union (IoU) (Wang et al., 2020) or related overlap-based measures that quantify agreement between predicted anomaly regions and ground-truth masks. Additionally, metrics such as Area Under the Per-Region

Overlap curve (AUPRO) aggregate localization performance across thresholds to emphasize region-level detection rather than isolated pixel accuracy.

While these metrics incorporate spatial overlap, they remain highly sensitive to thresholding choices and boundary alignment. Small variations along region boundaries can disproportionately affect scores, and fragmented predictions covering the correct area may be evaluated similarly to a single cohesive region. More broadly, existing localization metrics lack explicit mechanisms to account for structural properties such as contiguity, redundancy of predicted regions, or the distinction between a single consolidated anomaly and multiple disjoint detections.

**Operational Metrics and the Point-of-Decision** While threshold-independent curves are useful for model selection, industrial deployment necessitates a specific operating point. Consequently, threshold-dependent metrics such as Precision, Recall, and the F1-score remain the primary tools for quantifying performance in real-world inspection cycles. These measures provide a direct snapshot of a system's reliability: Recall ensures defect coverage, while Precision governs the rate of false alarms that could lead to unnecessary downtime. However, as these metrics are typically calculated pixel-wise, they lack the structural awareness necessary to distinguish between a single concise localization and multiple redundant predictions covering the same region.

**From Empirical Evaluation to Formal Characterization** Recent work in Time-Series AD (Wagner et al., 2025) has highlighted the fundamental limitations of commonly used evaluation metrics by formally specifying desirable metric behaviors and proving that many popular metrics violate these requirements under simple counterexamples. By introducing verifiable properties and analyzing metrics through a theoretical lens, this line of work demonstrates that empirical benchmarking alone is insufficient to guarantee meaningful or consistent evaluations.

In contrast, evaluation in VAD remains largely empirical. Current VAD metrics are typically justified by convention rather than by formally defined requirements, and their implicit assumptions about spatial structure and visual coherence are rarely articulated or verified. As a consequence, different metrics may produce contradictory rankings while failing to reflect critical inspection-oriented criteria, such as penalizing fragmented predictions or redundant detections.

Our work bridges this gap by extending the property-based evaluation paradigm to VAD. We introduce a set of spatial and structural axioms that formalize essential requirements for visual anomaly evaluation, providing a principled framework for analyzing existing metrics and designing new ones

that align with the semantic and operational demands of visual inspection.

# 4. Formalization

This section formalizes the limitations identified in Section 1 by specifying explicit assumptions and verifiable properties that evaluation metrics for VAD are expected to satisfy. The goal is not to propose universal axioms, but to characterize metric behavior under a well-defined and practically relevant inspection regime.

## 4.1. Assumptions

We formalize the evaluation of VAD metrics under the following assumptions, which reflect standard practice in industrial visual inspection and define the scope of our analysis.

ASSUMPTION 1 (RELIABLE ANNOTATIONS) *The ground-truth segmentation masks are assumed to be accurate and unambiguous. Pixels labeled as anomalous correspond to true defects, while pixels labeled as normal correspond to defect-free regions.*

ASSUMPTION 2 (OBJECT-LEVEL ALARM SEMANTICS) *Each predicted connected component is treated as an independent prediction that triggers inspection. Predictions are acted upon as they are produced, without access to ground-truth information at inference time.*

ASSUMPTION 3 (SPARSE AND SEPARABLE DEFECTS) *Ground-truth anomalies are assumed to be spatially sparse and non-overlapping. Consequently, multiple predicted components associated with the same ground-truth anomaly are considered redundant.*

ASSUMPTION 4 (SCORE ORDERING CONVENTION) *Evaluation metrics are assumed to produce scalar scores where larger values indicate better AD performance.*

These assumptions define a well-posed evaluation regime for analyzing metric behavior and align with common industrial inspection scenarios.

## 4.2. Formal Problem Setting

As illustrated in Figure 3, we define the structural relationship between pixels using an $L1$-path. We call $\gamma\colon [|X|] \to X \subset \mathbb{N} \times \mathbb{N}$ an $L1$-path between $x, y \in X$ in $X$ ($x \xrightarrow[X]{\gamma} y$) if $\gamma(1) = x$, $\gamma(|X|) = y$, and $\forall i \in [|X| - 1]\colon ||\gamma(i) - \gamma(i+1)||_1 \leq 1 \wedge \gamma(i) \in X$.

Let $g, p\colon [h] \times [w] \to [c]$ represent the ground truth and predicted maps, respectively. We define $I_i(g)$ as the set of maximal connected components $i$:

$$I_i(g) = \{A \subset g^{-1}(i)\colon \forall a, a' \in A\colon$$
$$\exists a \xrightarrow[A]{\gamma} a' \wedge \forall a \in A, a' \in g^{-1}(i)\colon ||a - a'||_1 = 1 \Rightarrow a' \in A\}$$

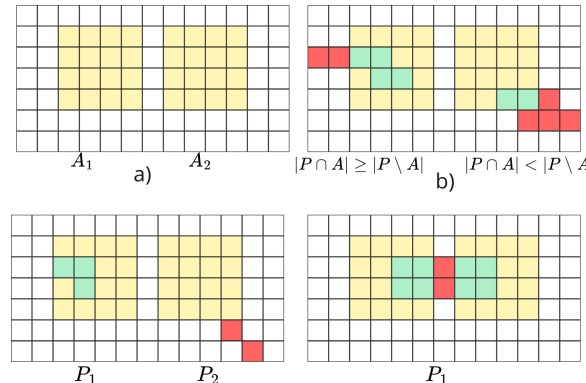

*Figure 3.* Criteria for $TD_i$. (a) Ground truth components $A_1$ and $A_2$ defined by $L1$-connectivity. (b) The Majority Overlap rule: the left prediction qualifies as a match ($|P \cap A| \geq |P \setminus A|$), while the right fails. (c) True Detection ($P_1$) vs. a False Positive component ($P_2$). (d) Detection failure: Prediction $P_1$ is disqualified because it bridges two distinct ground truth objects, violating the uniqueness constraint ($\forall A \neq A' \colon A' \cap P = \emptyset$).

Finally, a ground truth object $A$ is registered as a **True Detection** ($TD$) if there exists a predicted component $P \in I_i(p)$ that uniquely satisfies both the majority overlap and the separation constraints:

$$TD_i(p,g) = \{A \in I_i(g)\colon \exists P \in I_i(p)\colon$$
$$(|P \cap A| \geq |P \setminus A| \wedge \forall A \neq A' \in I_i(g)\colon A' \cap P = \emptyset)\}$$

With the formalization of object-level detections established, we now define a set of axiomatic properties that a robust visual anomaly detection metric $m(p,g)$ should satisfy. These properties–visually illustrated in Figure 4–ensure that the metric aligns with practical inspection requirements—such as penalizing fragmented detections and prioritizing central localization—rather than merely counting pixel-wise overlaps. In the following properties, we let $m(p,g)$ denote the score assigned to a prediction $p$ given ground truth $g$.

PROPERTY 1 (EXISTING PREDICTIONS) $A \in I_1(g)$, $TD_1(p,g) = TD_1(q,g) \sqcup \{A\}$, $p_{dom(p) \setminus A} = q_{dom(g) \setminus A}$, $q_A = 0$. Then $m(p,g) > m(q,g)$

This property requires that predictions that correctly detect an existing ground-truth anomaly receive a higher score than predictions that miss it entirely. A metric should therefore reward the presence of any valid detection over complete absence.

PROPERTY 2 (REDUNDANT TRUE DETECTIONS) $A \in TD_1(p,g)$, $P \in I_1(p)\colon |P \cap A| \geq |P \setminus A|$, $|I_1(p_A)| + 1 = |I_1(q_A)|$, $I \subset A \cap p^{-1}(0)$, $|I| \leq |P \cap A|$, $q = 1_I + p$, $\forall a \in P, a' \in I \nexists a \xrightarrow[dom(g)]{\gamma} a'$. Then $m(p,g) > m(q,g)$.

This property states that producing multiple redundant predictions for the same ground-truth anomaly should reduce the evaluation score. A single coherent detection is pre-ferred over fragmented or duplicated detections of the same anomaly.

PROPERTY 3 (MINIMIZE FALSE POSITIVES) $n \in g^{-1}(0)$, $p_{dom(g) \setminus \{n\}} = q_{dom(g) \setminus \{n\}}$, $p(n) = 0 \neq 1 = q(n)$. Then $m(p,g) > m(q,g)$.

This property enforces that introducing additional false-positive pixels in regions where no anomaly exists must decrease the metric score. Predictions should therefore avoid labeling normal regions as anomalous.

PROPERTY 4 (MINIMIZE NUMBER OF FALSE ALARMS) $p_{g=1} = q_{g=1}$, $I_1(q_{g=0}) = I_1(p_{g=0}) \sqcup A$. Then $m(p,g) > m(q,g)$.

This property requires that, for identical true-positive predictions, a method producing fewer disconnected false-alarm components should be scored higher. The metric should therefore penalize the number of false alarms, not only their total area.

PROPERTY 5 (DISTANCE-AWARE FALSE ALARM SEVERITY) $A_p \in I_1(p)$, $A_q \in I_1(q)$, $|A_p| = |A_q|$, $\forall A \in I_1(g)\colon A \cap A_p = A \cap A_q = \emptyset$, $I_1(p) \setminus A_p = I_1(q) \setminus A_q$. Let $d(A, I_1(g)) := \min_{a \in A, b \in \cup I_1(g)} \|a - b\|_1$. , if $d(A_p, I_1(g)) < d(A_q, I_1(g))$, then $m(p,g) > m(q,g)$.

False alarms occurring in the vicinity of a true anomaly are less detrimental than equally sized false alarms that appear far away from any anomalous region, as the latter are more likely to correspond to spurious detections and increase inspection effort.

PROPERTY 6 (TYPES OF FALSE ALARMS) $A \in TD_1(p,g)$, $P \in I_1(p)$, $|P \cap A| \geq |P \setminus A|$, $Q \in I_1(q)$, $(I_1(p) \setminus P) \cup (P \cap A) = I_1(q) \setminus Q$, $|P \setminus A| = |Q|$. Then $m(p,g) > m(q,g)$.

False-positive (FP) pixels are more disruptive when they form independent connected components than when they manifest as boundary imprecision. We define that for two predictions $p$ and $q$ with identical TP and FP pixel counts, $m(p,g) > m(q,g)$ if the FPs in $p$ are connected to a true detection while the FPs in $q$ form a separate component. This ensures that the metric penalizes the creation of additional, disjoint false detections.

PROPERTY 7 (PREDICTION SIZE) $A \in TD_1(p,g)$, $P \in I_1(p)\colon |P \cap A| \geq |P \setminus A|$, $a^* \in A \cap P$, $n^* \in P \setminus A$, $(I_1(p) \setminus P) \cup ((P \setminus \{a^*\}) \cup \{n^*\}) = I_1(q)$. Then $m(p,g) > m(q,g)$.

This property enforces that, for a correctly detected anomaly, removing true-positive pixels from an otherwise valid prediction should decrease the metric score. In other words, among predictions that detect the same ground-truth object, those that cover a larger portion of the anomaly should be preferred over those that only detect a smaller subset.

PROPERTY 8 (CENTRALITY OF TRUE DETECTIONS) *For $A \in I_1(g)$ and $a \in A$, we define $\delta_A(a) := \min_{b \in \partial A} \|a -$*

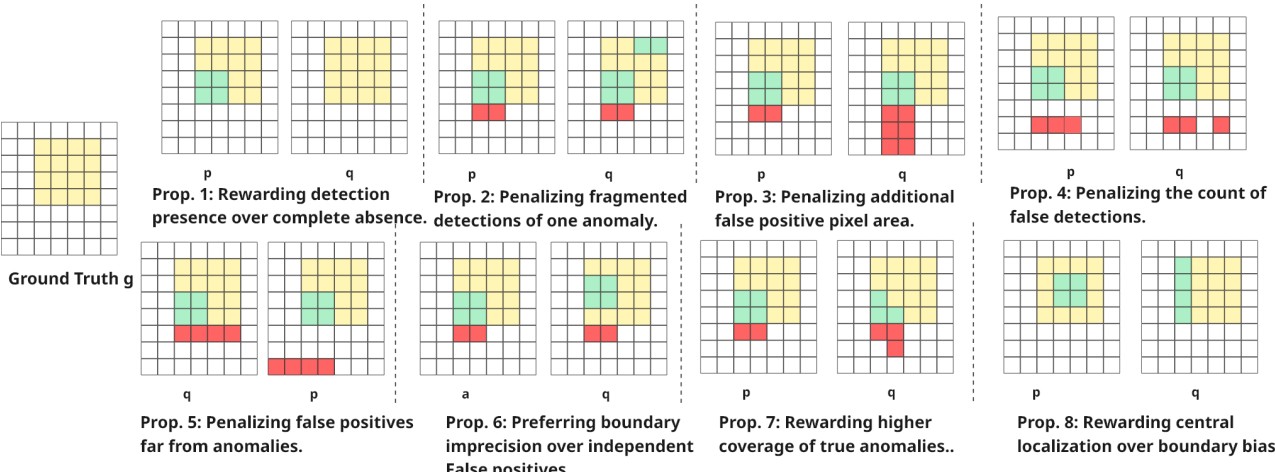

*Figure 4.* Visual illustration of the proposed framework properties. The **far-left panel** displays the reference Ground Truth ($g$). For each property (Prop. 1–8), we compare two candidate predictions, $p$ (left) and $q$ (right), against $g$. Green regions denote True Positives (TP), while red regions indicate False Positives (FP). Each property formalizes a spatial requirement for VAD metrics, such as penalizing fragmentation (Prop. 2) or rewarding central localization (Prop. 8).

$b\|_1$. $A \in TD_1(p, g)$, $P \in I_1(p)$, $Q \in I_1(q)$, $|P| = |Q|$, $P \subset A$, $Q \subset A$, $I_1(p) \setminus \{P\} = I_1(q) \setminus \{Q\}$,

$$\sum_{a \in P} \delta_A(a) > \sum_{a \in Q} \delta_A(a).$$

*Then* $m(p, g) > m(q, g)$.

This property enforces that, among equally sized detections fully contained within a ground-truth anomaly, predictions located more centrally should receive a higher score than those closer to the boundary. Central detections are typically more reliable, whereas boundary detections are more susceptible to localization noise.

## 5. Where Metrics Fail

In this section, we investigate six evaluation metrics with respect to the properties defined in Section 4. This axiomatic analysis serves to identify the structural limitations of current evaluation standards in the context of VAD.

As summarized in Table 1, our findings reveal that no single existing metric satisfies all eight desirable properties. For instance, while metrics like Accuracy and $F1$-score are sensitive to the total area of false positives (Property 3), they are entirely invariant to the number of disconnected false positives (Property 4) and the spatial centrality of detections (Property 8). Conversely, object-level metrics often fail to account for the distance-aware severity of false detections (Property 5), treating all spurious detections with equal weight regardless of their proximity to true anomalies.

This systemic gap underscores the need for a more comprehensive metric that aligns more closely with human inspection intuition and practical industrial requirements.

## 6. Towards Trustworthy Evaluation of VAD.

We extend the Advanced Alignment Accuracy Metric (ALARM) originally proposed for time-series AD (Wagner et al., 2025) to the image domain and introduce a new evaluation metric for VAD that satisfies all properties defined in Section 4. Our proposed metric, termed *Spatially Aware Anomaly Metric* (SAAM-ALARM), is designed to jointly reward accurate, spatially coherent detections while penalizing redundant and spurious predictions in a principled manner.

Let $I \subset \mathbb{Z}^2$ denote the image domain of size $H \times W$. The ground truth is represented as a set of $N_G$ disjoint connected components $\mathcal{A} = \{A_1, A_2, \ldots, A_{N_G}\}$, where each $A_i \subset I$ corresponds to an anomalous region. Similarly, the model prediction is defined as a set of $N_P$ disjoint connected components $\mathcal{P} = \{P_1, P_2, \ldots, P_{N_P}\}$.

At a high level, SAAM-ALARM computes an object-level reward for each ground-truth anomaly based on spatially weighted overlap with the best-matching prediction, while applying explicit penalties for redundant detections and false positive regions. To ensure that a single correct detection is not penalized, redundancy is discounted only for detections beyond the first. Finally, we normalize the score by the maximum attainable reward so that SAAM-ALARM lies in $[-1, 1]$ in the absence of penalties. The final score is defined as:

$$\text{SAAM-ALARM} = \tanh\left(\frac{1}{2}\left[\frac{1}{N_G}\sum_{i=1}^{N_G}\left(\frac{\alpha_i + 1}{b^{\max(0, n_i - 1)}}\right) - \sum_{P_j \in \mathcal{F}} \omega_j \cdot (\lambda + \beta(P_j))\right]\right)$$

*Table 1.* Comparison of metrics across P1–P8.

| Metric | P1 | P2 | P3 | P4 | P5 | P6 | P7 | P8 |
|---|---|---|---|---|---|---|---|---|
| Precision (Heckler-Kram et al., 2026) | × | × | × | × | × | × | ✓ | × |
| Recall (Heckler-Kram et al., 2026) | ✓ | × | × | × | × | × | ✓ | × |
| F1-score (Wang et al., 2020) | ✓ | × | × | × | × | × | ✓ | × |
| IOU (Wang et al., 2020) | ✓ | × | × | × | × | × | ✓ | × |
| CA (Rački et al., 2022) | ✓ | × | ✓ | × | × | × | ✓ | × |
| MCC (Krassnig & Gruber, 2026) | × | × | ✓ | × | × | × | × | × |

where $n_i = |\mathcal{C}_i|$ denotes the number of candidate predictions associated with $A_i$.

**Component Formalization** We now describe the four individual components of the proposed SAAM-ALARM metric, each designed to address a specific structural or operational requirement identified in the preceding axiomatic analysis.

*1. Spatially Modulated Object-Level Reward ($\alpha_i$)*
Each ground-truth component $A_i \in \mathcal{A}$ is modeled as a graph $G_i = (V_i, E_i)$, where pixels correspond to vertices and edges connect neighboring pixels. We define a central pixel $c_i \in A_i$ as the vertex with minimum eccentricity. A spatial weighting function $W_i(u)$ assigns higher importance to pixels closer to the center:

$$W_i(u) = \frac{\exp\left(-\frac{\|u - c_i\|^2}{2\sigma^2}\right)}{\sum_{v \in A_i} \exp\left(-\frac{\|v - c_i\|^2}{2\sigma^2}\right)}.$$

Let $P_i^* \in \mathcal{P}$ denote the prediction that maximizes overlap with $A_i$. The modulated reward is defined as:

$$\alpha_i = \text{IoU}(A_i, P_i^*) \cdot \sum_{u \in A_i \cap P_i^*} W_i(u),$$

which jointly captures spatial accuracy and region coverage.

*2. Redundancy Regularization ($b^{\max(0, n_i - 1)}$)*
To penalize redundant detections of the same anomaly, we define the set of candidate predictions for $A_i$ as

$$\mathcal{C}_i = \left\{ P_j \in \mathcal{P} \;\middle|\; \frac{|A_i \cap P_j|}{|P_j|} \geq \tau \right\}.$$

The redundancy factor is given by $n_i = |\mathcal{C}_i|$, and the reward for $A_i$ is exponentially discounted by $b^{n_i}$ with base $b > 1$. Larger values of $b$ impose stronger penalties on redundant detections, while $b = 2$ provides a balanced trade-off between redundancy suppression and reward preservation.

*3. Proximity-Weighted False Positive Penalty*
Let $\mathcal{F} \subset \mathcal{P}$ denote the set of false positive (unmatched) components. For each $P_j \in \mathcal{F}$, we define a proximity weight:

$$\omega_j = 1 - \exp\left(-\frac{d(P_j, \mathcal{A})}{k}\right),$$

where

$$d(P_j, \mathcal{A}) = \min_{u \in P_j, v \in \cup \mathcal{A}} \|u - v\|$$

is the minimum Euclidean distance to the nearest true anomaly. The parameter $k$ controls the saturation scale of the distance-based penalty. This formulation increases penalties for false detections located far from ground-truth anomalies.

*4. Dual Penalty Factors ($\lambda$, $\beta$)*
Each false positive component incurs a penalty composed of a fixed count term $\lambda$ and an area-aware size term $\beta(P_j)$:

$$\beta(P_j) = \frac{|P_j|}{\sum_{i=1}^{N_G} |A_i|},$$

where $\eta$ is a scaling coefficient. The parameter $\lambda$ balances the penalty contribution from false positive count against the positive reward components. This formulation penalizes false positive regions in proportion to their spatial footprint, while suppressing sensitivity to minor pixel-level artifacts.

**Hyperparameter Selection and Sensitivity** SAAM-ALARM includes three key hyperparameters: $b$, $k$, and $\lambda$. The parameter $b > 1$ controls the penalty applied to redundant detections, where larger values induce stronger exponential reward decay. We empirically select $b = 2$ as it effectively penalizes redundancy while preserving meaningful rewards for correct detections.

The parameter $k$ regulates the saturation behavior of the proximity-weighted false positive penalty. Small values lead to rapid saturation and reduced sensitivity to spatial distance, whereas excessively large values under-penalize distant false positives. We select $k = 10$ for stable and interpretable behavior. The weighting factor $\lambda = 0.5$ balances the contribution of false positives against the reward term.

Sensitivity analysis across representative anomaly categories demonstrates that the proposed metric behaves consistently across a reasonable range of parameter values. While absolute metric values vary slightly, the relative ranking of

methods remains largely preserved under moderate hyper-parameter perturbations.

**Theoretical guarantees.** The proposed SAAM-ALARM metric is designed to satisfy all axiomatic properties introduced in Section 4. Formal proofs that SAAM-ALARM satisfies Properties P1–P8 are provided in Appendix C.

# 7. Experiments

We evaluate the proposed framework and metric in two stages: (i) controlled case studies designed to expose specific structural limitations of conventional pixel-level metrics, and (ii) a comparative analysis of model rankings that demonstrates how these limitations affect practical conclusions.

## 7.1. Case Studies: Redundancy, False-Alarm Distance, and False-Alarm Count

To complement the axiomatic analysis, we present controlled case studies constructed from real MVTec-AD images and masks as shown in Fig. 5 (a). Each row isolates a single structural factor while keeping the overall pixel-level error profile nearly unchanged.

**Redundancy (top row).** Both predictions attain a precision $\approx 1.0$ and nearly identical recall/F1/IoU, since both lie entirely inside the ground-truth region. However, the right prediction produces two disconnected true-positive components (two alarms), whereas the left prediction produces a single coherent alarm. Pixel-level metrics treat both cases as essentially equivalent, while SAAM-ALARM assigns a substantially lower score to the redundant output (0.2500 vs. 0.5002), reflecting the additional inspection burden.

**Distance-aware false-alarm severity (middle row).** The two predictions have identical pixel-level statistics (precision/recall/F1/IoU/MCC), differing only in the location of the false-positive component. SAAM-ALARM assigns a lower score when the false alarm occurs farther from the ground truth (0.4492) than when it occurs close to it (0.4526), consistent with distance-aware penalization.

**Number of false alarms (bottom row).** Both predictions exhibit comparable pixel-level overlap, but the right prediction splits the false positives into two disconnected components, increasing the number of false alarms. While pixel-level metrics change only marginally, SAAM-ALARM decreases from 0.4517 to 0.4017 due to the increased alarm count. Overall, these examples demonstrate that pixel-wise overlap measures alone can be insensitive to practically relevant structural differences in anomaly predictions.

## 7.2. Evaluating Metric Consistency and Model Rankings.

To assess the practical impact of metric choice on comparative evaluation, we analyze the consistency of model rankings induced by different evaluation metrics. We evaluate six state-of-the-art AD models (UniAD (You et al., 2022), Desteg (Zhang et al., 2023), RD++ (Tien et al., 2023), Dinomaly (Guo et al., 2025), Vitad (Zhang et al., 2025), and P-MambaAD (Iqbal & Martinel, 2025)) on the MVTec AD dataset. Following the methodology of prior work (Wagner et al., 2025), we compare the rankings produced by our proposed SAAM-ALARM against traditional threshold-dependent AD metrics.

Our results highlight a significant divergence in model rankings across different evaluation criteria. As shown in Fig. 5 (b), ranking models by Recall would erroneously suggest that Dinomaly (0.96) and Pyramid-Mamba (0.93) are the best performing methods. However, even advanced balanced metrics fail to reach a consensus; for instance, Dest-Seg is ranked best by F1 (0.52) and MCC (0.56), while UniAD—our top performer under SAAM—is ranked significantly lower by those same measures (5th in F1 and 6th in Recall). This widespread ranking variance suggests that traditional metrics distort perception of performance because they ignore patterns such as redundancy and spatial latency.

In contrast, SAAM-ALARM yields a consistent and inspection-aligned ranking by explicitly accounting for alarm-level structure, including redundancy, false positive extent, and spatial distance. Under SAAM-ALARM, Dest-seg is ranked as the superior model (-0.24). Despite its lower recall (0.86) compared to Pyramid-Mamba, UniAD demonstrates higher reliability and concise localization, which are essential requirements for safety-critical or industrial applications.

These results suggest that traditional metrics distort performance perception by ignoring redundancy and spatial latency. This effect is further illustrated by specific failure cases (e.g., P-MambaAD), where dense and spatially dispersed false positives are not properly penalized under standard metrics. We provide a detailed analysis in the following subsection.

## 7.3. Failure modes of P-MambaAD.

Although P-MambaAD performs competitively under conventional pixel-wise metrics such as F1 and IoU, these metrics primarily capture overlap quality and do not explicitly account for redundancy or spatial dispersion of false positives. As a result, P-MambaAD exhibits failure modes that are not reflected in standard evaluation protocols.

In particular, we observe that P-MambaAD tends to produce more spatially dispersed and redundant detections compared

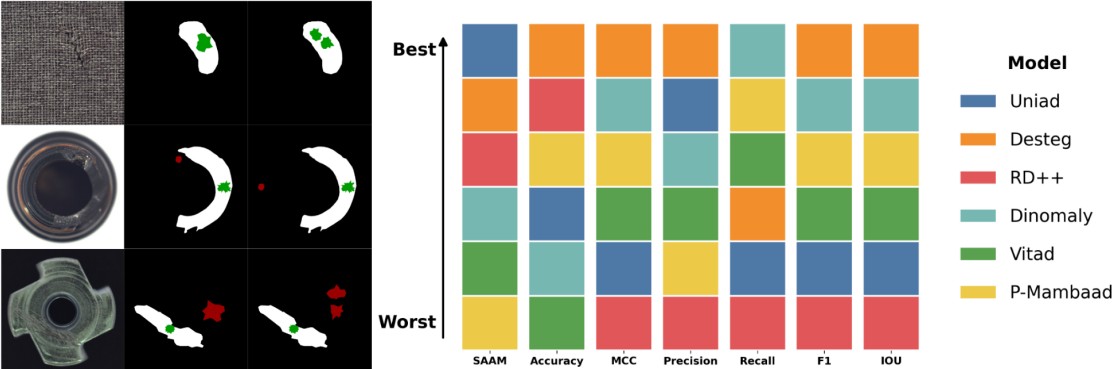

*Figure 5.* a) Real-data examples illustrating **structural failure modes** of pixel-level evaluation metrics. From top to bottom: redundant true detections, false positives near versus far from the ground truth, and increasing numbers of false alarms with similar pixel-level. b) **Model ranking variance on MVTec AD**. Significant color shuffling indicates that traditional metrics inconsistently rank models by failing to penalize redundancy. Our proposed SAAM (far left) provides a consistent ranking by accounting for spatial overlap and redundant predictions.

to other methods, leading to increased false positive density and less localized anomaly predictions. These characteristics are not adequately penalized by conventional metrics but are explicitly captured by SAAM-ALARM through its redundancy and distance-aware components, resulting in a lower overall ranking for P-MambaAD.

Detailed quantitative analysis further confirms higher redundancy rates and larger false positive spatial spread for P-MambaAD compared to stronger SAAM-ranked models.

### 7.4. Generalization to VisA Dataset

To further assess robustness beyond MVTec AD, we evaluate on the VisA (Zou et al., 2022) dataset, which contains more diverse and less structured anomalies. The inconsistencies of standard metrics persist across datasets and detectors. Models with similar Precision and Recall can differ significantly under SAAM-ALARM due to redundancy, spatially dispersed false positives, and localization structure. Unlike conventional metrics dominated by pixel overlap, SAAM-ALARM better reflects structural error patterns and yields more consistent rankings across UniAD, P-Mamba, and ViTAD. These findings show that the limitations observed on MVTec AD generalize to the more complex anomaly distributions of VisA. Additional results are provided in the Appendix.

## 8. Conclusion

We presented a property-based analysis of evaluation metrics for VAD, motivated by the observation that commonly used pixel-level measures often fail to capture practically relevant structural differences between predictions. We formalized a set of axiomatic properties that reflect key require-

ments in industrial visual inspection, including redundancy penalization, false-alarm control, and spatial awareness. Our analysis shows that widely adopted metrics such as precision, recall, F1-score, and IoU satisfy only a limited subset of these properties and can yield misleading assessments under realistic prediction patterns.

To address these limitations, we proposed SAAM-ALARM, a spatially aware evaluation metric that combines object-level rewards with explicit penalties for redundant and false alarms. We proved that SAAM-ALARM satisfies all proposed properties by construction and demonstrated, through synthetic examples and real-data case studies on MVTec-AD, that it distinguishes between predictions that are indistinguishable under pixel-level metrics. These results highlight the need for structurally informed evaluation and provide a principled foundation for more trustworthy assessment of VAD systems.

## Impact Statement

This paper presents work whose goal is to advance the field of Machine Learning. There are many potential societal consequences of our work, none which we feel must be specifically highlighted here.

## Acknowledgment

Supported by the DFG through FOR 5359 (ID 459419731), TRR 375 (ID 511263698), SPP 2298 (IDs 441826958 and 441826958), and SPP 2331 (IDs 441958259, 553345933, and 466468799), by the Carl-Zeiss Foundation through the initiative AI-Care, and by the BMFTR award 01IS24071A.

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

# A. Theorems.

**Theorem A.1.** *The **point-wise Precision***

$$Precision(g,p) = \frac{|p^{-1}(1) \cap g^{-1}(1)|}{|p^{-1}(1)|} \tag{1}$$

**Theorem A.2.** *The **point-wise Recall***

$$Recall(g,p) = \frac{|p^{-1}(1) \cap g^{-1}(1)|}{|g^{-1}(1)|} \tag{2}$$

**Theorem A.3.** *The **point-wise $F_1$-score***

$$F_1(g,p) = \frac{2|p^{-1}(1) \cap g^{-1}(1)|}{|p^{-1}(1)| + |g^{-1}(1)|} \tag{3}$$

**Theorem A.4.** *The **Intersection over Union (IOU)***

$$IoU(g,p) = \frac{|p^{-1}(1) \cap g^{-1}(1)|}{|p^{-1}(1) \cup g^{-1}(1)|} \tag{4}$$

**Theorem A.5.** *The **Classification Accuracy (CA)***

$$CA(g,p) = \frac{|(p^{-1}(1) \cap g^{-1}(1)) \cup (p^{-1}(0) \cap g^{-1}(0))|}{|p^{-1}(1) \cap g^{-1}(1)| + |p^{-1}(0) \cap g^{-1}(0)| + |p^{-1}(1) \cap g^{-1}(0)| + |p^{-1}(0) \cap g^{-1}(1)|}$$

$$= \frac{|(p^{-1}(1) \cap g^{-1}(1)) \cup (p^{-1}(0) \cap g^{-1}(0))|}{\Big(|p^{-1}(1) \cap g^{-1}(1)| + |p^{-1}(1) \cap g^{-1}(0)|\Big) + \Big(|p^{-1}(0) \cap g^{-1}(0)| + |p^{-1}(0) \cap g^{-1}(1)|\Big)} \tag{5}$$

$$= \frac{|(p^{-1}(1) \cap g^{-1}(1)) \cup (p^{-1}(0) \cap g^{-1}(0))|}{|p^{-1}(1)| + |p^{-1}(0)|}$$

**Theorem A.6.** *The **Matthews Correlation Coefficient (MCC)***

$$MCC(g,p) = \frac{|p^{-1}(1) \cap g^{-1}(1)| \cdot |p^{-1}(0) \cap g^{-1}(0)| - |p^{-1}(1) \cap g^{-1}(0)| \cdot |p^{-1}(0) \cap g^{-1}(1)|}{\sqrt{(|p^{-1}(1)| \cdot |p^{-1}(0)| \cdot |g^{-1}(1)| \cdot |g^{-1}(0)|)}} \tag{6}$$

# B. Proofs

## B.1. Theorem 1.1, The point-wise Precision.

**Prop.1** Consider the following example:

$$g = \begin{bmatrix} 0 & 0 & 0 & 0 & 0 & 1 & 1 & 0 \\ 0 & 0 & 0 & 0 & 0 & 1 & 1 & 0 \\ 0 & 0 & 0 & 0 & 0 & 0 & 0 & 0 \\ 0 & 0 & 0 & 0 & 0 & 0 & 0 & 0 \\ 0 & 0 & 1 & 1 & 0 & 0 & 0 & 0 \\ 0 & 0 & 1 & 1 & 0 & 0 & 0 & 0 \\ 0 & 0 & 0 & 0 & 0 & 0 & 0 & 0 \\ 0 & 0 & 0 & 0 & 0 & 0 & 0 & 0 \end{bmatrix}$$

$$p = \begin{bmatrix} 0 & 0 & 0 & 0 & 0 & 1 & 1 & 0 \\ 0 & 0 & 0 & 0 & 0 & 1 & 1 & 0 \\ 0 & 0 & 0 & 0 & 0 & 0 & 0 & 0 \\ 0 & 0 & 0 & 0 & 0 & 0 & 0 & 0 \\ 0 & 0 & 1 & 1 & 0 & 0 & 0 & 0 \\ 0 & 0 & 1 & 1 & 0 & 0 & 0 & 0 \\ 0 & 0 & 0 & 0 & 0 & 0 & 0 & 0 \\ 0 & 0 & 0 & 0 & 0 & 0 & 0 & 0 \end{bmatrix}$$

$$\hat{p} = \begin{bmatrix} 0 & 0 & 0 & 0 & 0 & 1 & 1 & 0 \\ 0 & 0 & 0 & 0 & 0 & 1 & 1 & 0 \\ 0 & 0 & 0 & 0 & 0 & 0 & 0 & 0 \\ 0 & 0 & 0 & 0 & 0 & 0 & 0 & 0 \\ 0 & 0 & 0 & 0 & 0 & 0 & 0 & 0 \\ 0 & 0 & 0 & 0 & 0 & 0 & 0 & 0 \\ 0 & 0 & 0 & 0 & 0 & 0 & 0 & 0 \\ 0 & 0 & 0 & 0 & 0 & 0 & 0 & 0 \end{bmatrix}$$

Then $Precision(g, p) = 1 = Precision(g, \hat{p})$.

**Prop.2** Consider the following example:

$$g = \begin{bmatrix} 0 & 0 & 0 & 0 & 0 & 0 & 0 & 0 \\ 0 & 1 & 1 & 1 & 1 & 1 & 0 & 0 \\ 0 & 1 & 1 & 1 & 1 & 1 & 0 & 0 \\ 0 & 1 & 1 & 1 & 1 & 1 & 0 & 0 \\ 0 & 1 & 1 & 1 & 1 & 1 & 0 & 0 \\ 0 & 1 & 1 & 1 & 1 & 1 & 0 & 0 \\ 0 & 0 & 0 & 0 & 0 & 0 & 0 & 0 \\ 0 & 0 & 0 & 0 & 0 & 0 & 0 & 0 \end{bmatrix}$$

$$p = \begin{bmatrix} 0 & 0 & 0 & 0 & 0 & 0 & 0 & 0 \\ 0 & 1 & 1 & 0 & 0 & 0 & 0 & 0 \\ 0 & 1 & 1 & 1 & 0 & 0 & 0 & 0 \\ 0 & 0 & 0 & 0 & 0 & 0 & 0 & 0 \\ 0 & 0 & 0 & 0 & 0 & 0 & 0 & 0 \\ 0 & 0 & 0 & 0 & 0 & 0 & 0 & 0 \\ 0 & 0 & 0 & 0 & 0 & 0 & 0 & 0 \\ 0 & 0 & 0 & 0 & 0 & 0 & 0 & 0 \end{bmatrix}$$

$$\hat{p} = \begin{bmatrix} 0 & 0 & 0 & 0 & 0 & 0 & 0 & 0 \\ 0 & 1 & 1 & 0 & 0 & 0 & 0 & 0 \\ 0 & 1 & 1 & 1 & 0 & 0 & 0 & 0 \\ 0 & 0 & 0 & 0 & 0 & 0 & 0 & 0 \\ 0 & 0 & 1 & 1 & 0 & 0 & 0 & 0 \\ 0 & 0 & 0 & 1 & 0 & 0 & 0 & 0 \\ 0 & 0 & 0 & 0 & 0 & 0 & 0 & 0 \\ 0 & 0 & 0 & 0 & 0 & 0 & 0 & 0 \end{bmatrix}$$

Then $Precision(g, p) = 1 = Precision(g, \hat{p})$.

**Prop.3** Consider the following example:

$$g = \begin{bmatrix} 0 & 0 & 0 & 0 & 0 & 0 & 1 & 0 \\ 0 & 0 & 0 & 0 & 0 & 1 & 1 & 0 \\ 0 & 0 & 0 & 0 & 0 & 0 & 0 & 0 \\ 0 & 0 & 0 & 0 & 0 & 0 & 0 & 0 \\ 0 & 0 & 0 & 0 & 0 & 0 & 0 & 0 \\ 0 & 0 & 0 & 0 & 0 & 0 & 0 & 0 \\ 0 & 0 & 0 & 0 & 0 & 0 & 0 & 0 \\ 0 & 0 & 0 & 0 & 0 & 0 & 0 & 0 \end{bmatrix}$$

$$
p = \begin{bmatrix}
0 & 0 & 0 & 0 & 0 & 0 & 0 & 0 \\
0 & 0 & 0 & 0 & 0 & 0 & 0 & 0 \\
0 & 0 & 0 & 0 & 0 & 0 & 0 & 0 \\
0 & 0 & 0 & 0 & 0 & 0 & 0 & 0 \\
0 & 0 & 1 & 1 & 0 & 0 & 0 & 0 \\
0 & 0 & 1 & 0 & 0 & 0 & 0 & 0 \\
0 & 0 & 0 & 0 & 0 & 0 & 0 & 0 \\
0 & 0 & 0 & 0 & 0 & 0 & 0 & 0
\end{bmatrix}
$$

$$
\hat{p} = \begin{bmatrix}
0 & 0 & 0 & 0 & 0 & 0 & 0 & 0 \\
0 & 0 & 0 & 0 & 0 & 0 & 0 & 0 \\
0 & 0 & 0 & 0 & 0 & 0 & 0 & 0 \\
0 & 0 & 0 & 0 & 0 & 0 & 0 & 0 \\
0 & 0 & 1 & 1 & 0 & 0 & 0 & 0 \\
0 & 0 & 1 & 1 & 0 & 0 & 0 & 0 \\
0 & 0 & 1 & 1 & 0 & 0 & 0 & 0 \\
0 & 0 & 0 & 0 & 0 & 0 & 0 & 0
\end{bmatrix}
$$

Then $Precision(g, p) = 0 = Precision(g, \hat{p})$.

**Prop.4** Consider the following example:

$$
g = \begin{bmatrix}
0 & 0 & 0 & 0 & 0 & 0 & 1 & 0 \\
0 & 0 & 0 & 0 & 0 & 1 & 1 & 0 \\
0 & 0 & 0 & 0 & 0 & 0 & 0 & 0 \\
0 & 0 & 0 & 0 & 0 & 0 & 0 & 0 \\
0 & 0 & 0 & 0 & 0 & 0 & 0 & 0 \\
0 & 0 & 0 & 0 & 0 & 0 & 0 & 0 \\
0 & 0 & 0 & 0 & 0 & 0 & 0 & 0 \\
0 & 0 & 0 & 0 & 0 & 0 & 0 & 0
\end{bmatrix}
$$

$$
p = \begin{bmatrix}
0 & 0 & 0 & 0 & 0 & 0 & 0 & 0 \\
0 & 0 & 0 & 0 & 0 & 0 & 0 & 0 \\
0 & 0 & 0 & 0 & 0 & 0 & 0 & 0 \\
0 & 0 & 0 & 0 & 0 & 0 & 0 & 0 \\
0 & 0 & 0 & 1 & 0 & 0 & 0 & 0 \\
0 & 0 & 0 & 0 & 0 & 0 & 0 & 0 \\
0 & 0 & 0 & 0 & 0 & 0 & 0 & 0 \\
0 & 0 & 0 & 0 & 0 & 0 & 0 & 0
\end{bmatrix}
$$

$$
\hat{p} = \begin{bmatrix}
0 & 0 & 0 & 0 & 0 & 0 & 0 & 0 \\
0 & 0 & 0 & 0 & 0 & 0 & 0 & 0 \\
0 & 0 & 0 & 0 & 0 & 0 & 0 & 0 \\
0 & 0 & 0 & 0 & 0 & 0 & 0 & 0 \\
0 & 0 & 1 & 1 & 0 & 0 & 0 & 0 \\
0 & 0 & 0 & 1 & 0 & 0 & 0 & 0 \\
0 & 0 & 0 & 0 & 0 & 0 & 0 & 0 \\
0 & 0 & 0 & 0 & 0 & 0 & 0 & 0
\end{bmatrix}
$$

Then $Precision(g, p) = 0 = Precision(g, \hat{p})$.

**Prop.5** Consider the following example:

$$g = \begin{bmatrix} 0 & 0 & 0 & 0 & 0 & 0 & 0 & 0 \\ 0 & 1 & 1 & 1 & 1 & 0 & 0 & 0 \\ 0 & 1 & 1 & 1 & 1 & 0 & 0 & 0 \\ 0 & 1 & 1 & 1 & 1 & 0 & 0 & 0 \\ 0 & 1 & 1 & 1 & 1 & 0 & 0 & 0 \\ 0 & 1 & 1 & 1 & 1 & 0 & 0 & 0 \\ 0 & 0 & 0 & 0 & 0 & 0 & 0 & 0 \\ 0 & 0 & 0 & 0 & 0 & 0 & 0 & 0 \end{bmatrix}$$

$$p = \begin{bmatrix} 0 & 0 & 0 & 0 & 0 & 1 & 1 & 0 \\ 0 & 0 & 0 & 0 & 0 & 1 & 1 & 0 \\ 0 & 0 & 0 & 0 & 0 & 0 & 0 & 0 \\ 0 & 0 & 1 & 0 & 0 & 0 & 0 & 0 \\ 0 & 0 & 1 & 1 & 0 & 0 & 0 & 0 \\ 0 & 0 & 1 & 1 & 0 & 0 & 0 & 0 \\ 0 & 0 & 0 & 0 & 0 & 0 & 1 & 1 \\ 0 & 0 & 0 & 0 & 0 & 0 & 1 & 0 \end{bmatrix}$$

$$\hat{p} = \begin{bmatrix} 0 & 0 & 0 & 0 & 0 & 0 & 0 & 0 \\ 0 & 0 & 0 & 0 & 0 & 0 & 0 & 0 \\ 0 & 0 & 0 & 0 & 0 & 0 & 0 & 0 \\ 0 & 0 & 1 & 0 & 0 & 0 & 0 & 0 \\ 0 & 0 & 1 & 1 & 0 & 0 & 0 & 0 \\ 0 & 0 & 1 & 1 & 0 & 0 & 0 & 0 \\ 0 & 0 & 0 & 0 & 0 & 0 & 1 & 1 \\ 0 & 0 & 0 & 0 & 0 & 0 & 1 & 0 \end{bmatrix}$$

Then $Precision(g, p) = 5/8 = Precision(g, \hat{p})$.

**Prop.6** Consider the following example.

$$g = \begin{bmatrix} 0 & 0 & 0 & 0 & 0 & 0 & 0 & 0 \\ 0 & 1 & 1 & 1 & 1 & 0 & 0 & 0 \\ 0 & 1 & 1 & 1 & 1 & 0 & 0 & 0 \\ 0 & 1 & 1 & 1 & 1 & 0 & 0 & 0 \\ 0 & 1 & 1 & 1 & 1 & 0 & 0 & 0 \\ 0 & 1 & 1 & 1 & 1 & 0 & 0 & 0 \\ 0 & 0 & 0 & 0 & 0 & 0 & 0 & 0 \\ 0 & 0 & 0 & 0 & 0 & 0 & 0 & 0 \end{bmatrix}$$

$$p = \begin{bmatrix} 0 & 0 & 0 & 0 & 0 & 1 & 1 & 0 \\ 0 & 0 & 0 & 0 & 0 & 1 & 1 & 0 \\ 0 & 0 & 0 & 0 & 0 & 0 & 0 & 0 \\ 0 & 0 & 0 & 0 & 0 & 0 & 0 & 0 \\ 0 & 0 & 1 & 1 & 1 & 0 & 0 & 0 \\ 0 & 0 & 1 & 1 & 1 & 0 & 0 & 0 \\ 0 & 0 & 1 & 1 & 0 & 0 & 0 & 0 \\ 0 & 0 & 0 & 0 & 0 & 0 & 0 & 0 \end{bmatrix}$$

$$\hat{p} = \begin{bmatrix} 0 & 0 & 0 & 0 & 0 & 1 & 1 & 0 \\ 0 & 0 & 0 & 0 & 0 & 1 & 1 & 0 \\ 0 & 0 & 0 & 0 & 0 & 0 & 0 & 0 \\ 0 & 0 & 0 & 0 & 0 & 0 & 0 & 0 \\ 0 & 0 & 1 & 1 & 1 & 0 & 0 & 0 \\ 0 & 0 & 1 & 1 & 1 & 0 & 0 & 0 \\ 0 & 0 & 0 & 0 & 0 & 0 & 0 & 0 \\ 0 & 0 & 1 & 1 & 0 & 0 & 0 & 0 \end{bmatrix}$$

Then $Precision(g,p) = 3/4 = Precision(g,\hat{p})$.

**Prop.7** By construction, $a^* \in p^{-1}(1) \cap g^{-1}(1)$, hence removing $a^*$ decreases the number of true-positive pixels by one. Also, $n^* \in g^{-1}(0)$, so adding $n^*$ does not increase the true-positive set. Therefore,

$$|q^{-1}(1) \cap g^{-1}(1)| = |p^{-1}(1) \cap g^{-1}(1)| - 1.$$

Moreover, the shift is size-preserving, so

$$|q^{-1}(1)| = |p^{-1}(1) \setminus \{a^*\}| + 1 = |p^{-1}(1)|.$$

Hence,

$$\text{Precision}(g,q) = \frac{|q^{-1}(1) \cap g^{-1}(1)|}{|q^{-1}(1)|} = \frac{|p^{-1}(1) \cap g^{-1}(1)| - 1}{|p^{-1}(1)|} < \frac{|p^{-1}(1) \cap g^{-1}(1)|}{|p^{-1}(1)|} = \text{Precision}(g,p).$$

**Prop.8** Consider the following example:

$$g = \begin{bmatrix} 0 & 0 & 0 & 0 & 0 & 0 & 0 & 0 \\ 0 & 1 & 1 & 1 & 1 & 1 & 0 & 0 \\ 0 & 1 & 1 & 1 & 1 & 1 & 0 & 0 \\ 0 & 1 & 1 & 1 & 1 & 1 & 0 & 0 \\ 0 & 1 & 1 & 1 & 1 & 1 & 0 & 0 \\ 0 & 1 & 1 & 1 & 1 & 1 & 0 & 0 \\ 0 & 0 & 0 & 0 & 0 & 0 & 0 & 0 \\ 0 & 0 & 0 & 0 & 0 & 0 & 0 & 0 \end{bmatrix}$$

$$p = \begin{bmatrix} 0 & 0 & 0 & 0 & 0 & 0 & 0 & 0 \\ 0 & 0 & 0 & 0 & 0 & 0 & 0 & 0 \\ 0 & 0 & 1 & 1 & 1 & 0 & 0 & 0 \\ 0 & 0 & 1 & 1 & 1 & 0 & 0 & 0 \\ 0 & 0 & 0 & 0 & 0 & 0 & 0 & 0 \\ 0 & 0 & 0 & 0 & 0 & 0 & 0 & 0 \\ 0 & 0 & 0 & 0 & 0 & 0 & 0 & 0 \\ 0 & 0 & 0 & 0 & 0 & 0 & 0 & 0 \end{bmatrix}$$

$$\hat{p} = \begin{bmatrix} 0 & 0 & 0 & 0 & 0 & 0 & 0 & 0 \\ 0 & 1 & 1 & 0 & 0 & 0 & 0 & 0 \\ 0 & 1 & 0 & 0 & 0 & 0 & 0 & 0 \\ 0 & 1 & 0 & 0 & 0 & 0 & 0 & 0 \\ 0 & 1 & 0 & 0 & 0 & 0 & 0 & 0 \\ 0 & 0 & 0 & 0 & 0 & 0 & 0 & 0 \\ 0 & 0 & 0 & 0 & 0 & 0 & 0 & 0 \\ 0 & 0 & 0 & 0 & 0 & 0 & 0 & 0 \end{bmatrix}$$

Then $Precision(g,p) = 1 = Precision(g,\hat{p})$.

## B.2. Theorem 3.2, The point-wise Recall.

**Prop.1** Let $W \in A(g)$ and let $p, \hat{p}$ be two predictions such that

$$p_{[n \times n] \setminus W} = \hat{p}_{[n \times n] \setminus W}, \qquad p_W = 0, \qquad |\hat{p}^{-1}(1) \cap W| > 0.$$

Then

$$\mathrm{Recall}(g, \hat{p}) > \mathrm{Recall}(g, p).$$

Since $p$ and $\hat{p}$ coincide on $[n \times n] \setminus W$, we have

$$|p^{-1}(1) \cap g^{-1}(1) \cap ([n \times n] \setminus W)| = |\hat{p}^{-1}(1) \cap g^{-1}(1) \cap ([n \times n] \setminus W)|.$$

Moreover, $p_W = 0$ implies $p^{-1}(1) \cap W = \emptyset$, hence

$$|p^{-1}(1) \cap g^{-1}(1) \cap W| = 0.$$

Since $W \in A(g)$, we have $W \subseteq g^{-1}(1)$, and $|\hat{p}^{-1}(1) \cap W| > 0$ yields

$$|\hat{p}^{-1}(1) \cap g^{-1}(1) \cap W| = |\hat{p}^{-1}(1) \cap W| > 0.$$

Therefore,

$$|\hat{p}^{-1}(1) \cap g^{-1}(1)| > |p^{-1}(1) \cap g^{-1}(1)|.$$

The denominator $|g^{-1}(1)|$ is the same for both predictions, so

$$\mathrm{Recall}(g, \hat{p}) = \frac{|\hat{p}^{-1}(1) \cap g^{-1}(1)|}{|g^{-1}(1)|} > \frac{|p^{-1}(1) \cap g^{-1}(1)|}{|g^{-1}(1)|} = \mathrm{Recall}(g, p).$$

**Prop.2** Consider the following example:

$$g = \begin{bmatrix} 0 & 0 & 0 & 0 & 0 & 0 & 0 & 0 \\ 0 & 1 & 1 & 1 & 1 & 1 & 0 & 0 \\ 0 & 1 & 1 & 1 & 1 & 1 & 0 & 0 \\ 0 & 1 & 1 & 1 & 1 & 1 & 0 & 0 \\ 0 & 1 & 1 & 1 & 1 & 1 & 0 & 0 \\ 0 & 1 & 1 & 1 & 1 & 1 & 0 & 0 \\ 0 & 0 & 0 & 0 & 0 & 0 & 0 & 0 \\ 0 & 0 & 0 & 0 & 0 & 0 & 0 & 0 \end{bmatrix}$$

$$p = \begin{bmatrix} 0 & 0 & 0 & 0 & 0 & 0 & 0 & 0 \\ 0 & 1 & 1 & 1 & 0 & 0 & 0 & 0 \\ 0 & 1 & 1 & 1 & 0 & 0 & 0 & 0 \\ 0 & 0 & 0 & 0 & 0 & 0 & 0 & 0 \\ 0 & 0 & 0 & 0 & 0 & 0 & 0 & 0 \\ 0 & 0 & 0 & 0 & 0 & 0 & 0 & 0 \\ 0 & 0 & 0 & 0 & 0 & 0 & 0 & 0 \\ 0 & 0 & 0 & 0 & 0 & 0 & 0 & 0 \end{bmatrix}$$

$$\hat{p} = \begin{bmatrix} 0 & 0 & 0 & 0 & 0 & 0 & 0 & 0 \\ 0 & 1 & 1 & 1 & 0 & 0 & 0 & 0 \\ 0 & 1 & 1 & 1 & 0 & 0 & 0 & 0 \\ 0 & 0 & 0 & 0 & 0 & 0 & 0 & 0 \\ 0 & 0 & 1 & 1 & 0 & 0 & 0 & 0 \\ 0 & 0 & 0 & 1 & 0 & 0 & 0 & 0 \\ 0 & 0 & 0 & 0 & 0 & 0 & 0 & 0 \\ 0 & 0 & 0 & 0 & 0 & 0 & 0 & 0 \end{bmatrix}$$

Then $Recall(g, p) = 6/25 < 9/25 = Recall(g, \hat{p})$.

**Prop.3** Consider the following example:

$$g = \begin{bmatrix} 0 & 0 & 0 & 1 & 1 & 0 & 0 & 0 \\ 0 & 0 & 0 & 1 & 1 & 0 & 0 & 0 \\ 0 & 0 & 0 & 0 & 1 & 0 & 0 & 0 \\ 0 & 0 & 0 & 0 & 0 & 0 & 0 & 0 \\ 0 & 0 & 0 & 0 & 0 & 0 & 0 & 0 \\ 0 & 0 & 0 & 0 & 0 & 0 & 0 & 0 \\ 0 & 0 & 0 & 0 & 0 & 0 & 0 & 0 \\ 0 & 0 & 0 & 0 & 0 & 0 & 0 & 0 \end{bmatrix}$$

$$p = \begin{bmatrix} 0 & 0 & 0 & 1 & 1 & 0 & 0 & 0 \\ 0 & 0 & 0 & 1 & 1 & 0 & 0 & 0 \\ 0 & 0 & 0 & 0 & 1 & 0 & 0 & 0 \\ 0 & 0 & 0 & 0 & 0 & 0 & 0 & 0 \\ 0 & 0 & 0 & 0 & 0 & 0 & 0 & 0 \\ 0 & 0 & 0 & 1 & 1 & 0 & 0 & 0 \\ 0 & 0 & 0 & 0 & 0 & 0 & 0 & 0 \\ 0 & 0 & 0 & 0 & 0 & 0 & 0 & 0 \end{bmatrix}$$

$$\hat{p} = \begin{bmatrix} 0 & 0 & 0 & 1 & 1 & 0 & 0 & 0 \\ 0 & 0 & 0 & 1 & 1 & 0 & 0 & 0 \\ 0 & 0 & 0 & 0 & 1 & 0 & 0 & 0 \\ 0 & 0 & 0 & 0 & 0 & 0 & 0 & 0 \\ 0 & 0 & 0 & 0 & 0 & 0 & 0 & 0 \\ 0 & 0 & 0 & 1 & 1 & 0 & 0 & 0 \\ 0 & 0 & 0 & 1 & 1 & 0 & 0 & 0 \\ 0 & 0 & 0 & 0 & 0 & 0 & 0 & 0 \end{bmatrix}$$

Then $Recall(g, p) = 1 \ Recall(g, \hat{p})$.

**Prop.4** Consider the following example:

$$g = \begin{bmatrix} 0 & 0 & 0 & 1 & 1 & 0 & 0 & 0 \\ 0 & 0 & 0 & 1 & 1 & 0 & 0 & 0 \\ 0 & 0 & 0 & 0 & 1 & 0 & 0 & 0 \\ 0 & 0 & 0 & 0 & 0 & 0 & 0 & 0 \\ 0 & 0 & 0 & 0 & 0 & 0 & 0 & 0 \\ 0 & 0 & 0 & 0 & 0 & 0 & 0 & 0 \\ 0 & 0 & 0 & 0 & 0 & 0 & 0 & 0 \\ 0 & 0 & 0 & 0 & 0 & 0 & 0 & 0 \end{bmatrix}$$

$$p = \begin{bmatrix} 0 & 0 & 0 & 1 & 1 & 0 & 0 & 0 \\ 0 & 0 & 0 & 1 & 1 & 0 & 0 & 0 \\ 0 & 0 & 0 & 0 & 1 & 0 & 0 & 0 \\ 0 & 0 & 0 & 0 & 0 & 0 & 0 & 0 \\ 0 & 0 & 0 & 0 & 0 & 0 & 0 & 0 \\ 0 & 0 & 0 & 1 & 1 & 0 & 0 & 0 \\ 0 & 0 & 0 & 0 & 0 & 0 & 0 & 0 \\ 0 & 0 & 0 & 0 & 0 & 0 & 0 & 0 \end{bmatrix}$$

$$\hat{p} = \begin{bmatrix} 0 & 0 & 0 & 1 & 1 & 0 & 0 & 0 \\ 0 & 0 & 0 & 1 & 1 & 0 & 0 & 0 \\ 0 & 0 & 0 & 0 & 1 & 0 & 0 & 0 \\ 0 & 0 & 0 & 0 & 0 & 0 & 0 & 0 \\ 1 & 1 & 0 & 0 & 0 & 0 & 1 & 0 \\ 1 & 1 & 0 & 1 & 1 & 0 & 1 & 0 \\ 0 & 0 & 0 & 1 & 1 & 0 & 0 & 0 \\ 0 & 0 & 0 & 0 & 0 & 0 & 0 & 0 \end{bmatrix}$$

Then $Recall(g, p) = 1$ $Recall(g, \hat{p})$.

**Prop.5** Consider the following example:

$$g = \begin{bmatrix} 0 & 0 & 0 & 0 & 0 & 0 & 0 & 0 \\ 0 & 1 & 1 & 1 & 1 & 0 & 0 & 0 \\ 0 & 1 & 1 & 1 & 1 & 0 & 0 & 0 \\ 0 & 1 & 1 & 1 & 1 & 0 & 0 & 0 \\ 0 & 1 & 1 & 1 & 1 & 0 & 0 & 0 \\ 0 & 1 & 1 & 1 & 1 & 0 & 0 & 0 \\ 0 & 0 & 0 & 0 & 0 & 0 & 0 & 0 \\ 0 & 0 & 0 & 0 & 0 & 0 & 0 & 0 \end{bmatrix}$$

$$p = \begin{bmatrix} 0 & 0 & 0 & 0 & 0 & 1 & 1 & 0 \\ 0 & 0 & 0 & 0 & 0 & 1 & 1 & 0 \\ 0 & 0 & 0 & 0 & 0 & 0 & 0 & 0 \\ 0 & 0 & 1 & 0 & 0 & 0 & 0 & 0 \\ 0 & 0 & 1 & 1 & 0 & 0 & 0 & 0 \\ 0 & 0 & 1 & 1 & 0 & 0 & 0 & 0 \\ 0 & 0 & 0 & 0 & 0 & 0 & 1 & 1 \\ 0 & 0 & 0 & 0 & 0 & 0 & 1 & 0 \end{bmatrix}$$

$$\hat{p} = \begin{bmatrix} 0 & 0 & 0 & 0 & 0 & 0 & 0 & 0 \\ 0 & 0 & 0 & 0 & 0 & 0 & 0 & 0 \\ 0 & 0 & 0 & 0 & 0 & 0 & 0 & 0 \\ 0 & 0 & 1 & 0 & 0 & 0 & 0 & 0 \\ 0 & 0 & 1 & 1 & 0 & 0 & 0 & 0 \\ 0 & 0 & 1 & 1 & 0 & 0 & 0 & 0 \\ 0 & 0 & 0 & 0 & 0 & 0 & 1 & 1 \\ 0 & 0 & 0 & 0 & 0 & 0 & 1 & 0 \end{bmatrix}$$

Then $Recall(g, p) = 1/5 = Recall(g, \hat{p})$.

**Prop.6** Consider the following example.

$$g = \begin{bmatrix} 0 & 0 & 0 & 0 & 0 & 0 & 0 & 0 \\ 0 & 1 & 1 & 1 & 1 & 0 & 0 & 0 \\ 0 & 1 & 1 & 1 & 1 & 0 & 0 & 0 \\ 0 & 1 & 1 & 1 & 1 & 0 & 0 & 0 \\ 0 & 1 & 1 & 1 & 1 & 0 & 0 & 0 \\ 0 & 1 & 1 & 1 & 1 & 0 & 0 & 0 \\ 0 & 0 & 0 & 0 & 0 & 0 & 0 & 0 \\ 0 & 0 & 0 & 0 & 0 & 0 & 0 & 0 \end{bmatrix}$$

$$p = \begin{bmatrix} 0 & 0 & 0 & 0 & 0 & 1 & 1 & 0 \\ 0 & 0 & 0 & 0 & 0 & 1 & 1 & 0 \\ 0 & 0 & 0 & 0 & 0 & 0 & 0 & 0 \\ 0 & 0 & 0 & 0 & 0 & 0 & 0 & 0 \\ 0 & 0 & 1 & 1 & 1 & 0 & 0 & 0 \\ 0 & 0 & 1 & 1 & 1 & 0 & 0 & 0 \\ 0 & 0 & 1 & 1 & 0 & 0 & 0 & 0 \\ 0 & 0 & 0 & 0 & 0 & 0 & 0 & 0 \end{bmatrix}$$

$$\hat{p} = \begin{bmatrix} 0 & 0 & 0 & 0 & 0 & 1 & 1 & 0 \\ 0 & 0 & 0 & 0 & 0 & 1 & 1 & 0 \\ 0 & 0 & 0 & 0 & 0 & 0 & 0 & 0 \\ 0 & 0 & 0 & 0 & 0 & 0 & 0 & 0 \\ 0 & 0 & 1 & 1 & 1 & 0 & 0 & 0 \\ 0 & 0 & 1 & 1 & 1 & 0 & 0 & 0 \\ 0 & 0 & 0 & 0 & 0 & 0 & 0 & 0 \\ 0 & 0 & 1 & 1 & 0 & 0 & 0 & 0 \end{bmatrix}$$

Then $Recall(g, p) = 1/4 = Recall(g, \hat{p})$.

**Prop.7** Since $a^* \in p^{-1}(1) \cap g^{-1}(1)$, removing $a^*$ decreases the number of true-positive pixels by one. Moreover, $n^* \in g^{-1}(0)$ implies that adding $n^*$ does not increase the true-positive set. Hence,

$$|q^{-1}(1) \cap g^{-1}(1)| = |p^{-1}(1) \cap g^{-1}(1)| - 1.$$

The denominator $|g^{-1}(1)|$ is unchanged. Therefore,

$$\text{Recall}(g, q) = \frac{|p^{-1}(1) \cap g^{-1}(1)| - 1}{|g^{-1}(1)|} < \frac{|p^{-1}(1) \cap g^{-1}(1)|}{|g^{-1}(1)|} = \text{Recall}(g, p),$$

**Prop.8** Consider the following example:

$$g = \begin{bmatrix} 0 & 0 & 0 & 0 & 0 & 0 & 0 & 0 \\ 0 & 1 & 1 & 1 & 1 & 1 & 0 & 0 \\ 0 & 1 & 1 & 1 & 1 & 1 & 0 & 0 \\ 0 & 1 & 1 & 1 & 1 & 1 & 0 & 0 \\ 0 & 1 & 1 & 1 & 1 & 1 & 0 & 0 \\ 0 & 1 & 1 & 1 & 1 & 1 & 0 & 0 \\ 0 & 0 & 0 & 0 & 0 & 0 & 0 & 0 \\ 0 & 0 & 0 & 0 & 0 & 0 & 0 & 0 \end{bmatrix}$$

$$p = \begin{bmatrix} 0 & 0 & 0 & 0 & 0 & 0 & 0 & 0 \\ 0 & 0 & 0 & 0 & 0 & 0 & 0 & 0 \\ 0 & 0 & 1 & 1 & 1 & 0 & 0 & 0 \\ 0 & 0 & 1 & 1 & 1 & 0 & 0 & 0 \\ 0 & 0 & 0 & 0 & 0 & 0 & 0 & 0 \\ 0 & 0 & 0 & 0 & 0 & 0 & 0 & 0 \\ 0 & 0 & 0 & 0 & 0 & 0 & 0 & 0 \\ 0 & 0 & 0 & 0 & 0 & 0 & 0 & 0 \end{bmatrix}$$

$$\hat{p} = \begin{bmatrix} 0 & 0 & 0 & 0 & 0 & 0 & 0 & 0 \\ 0 & 1 & 1 & 0 & 0 & 0 & 0 & 0 \\ 0 & 1 & 0 & 0 & 0 & 0 & 0 & 0 \\ 0 & 1 & 0 & 0 & 0 & 0 & 0 & 0 \\ 0 & 1 & 0 & 0 & 0 & 0 & 0 & 0 \\ 0 & 0 & 0 & 0 & 0 & 0 & 0 & 0 \\ 0 & 0 & 0 & 0 & 0 & 0 & 0 & 0 \\ 0 & 0 & 0 & 0 & 0 & 0 & 0 & 0 \end{bmatrix}$$

Then $Recall(g, p) = 1/2 = Recall(g, \hat{p})$.

### B.3. Theorem 3.3, The point-wise $F1 - score$.

**Prop.1** Since $p_W = 0$, we have $p^{-1}(1) \cap W = \emptyset$, hence

$$|p^{-1}(1) \cap g^{-1}(1) \cap W| = 0.$$

Let

$$\epsilon := |\hat{p}^{-1}(1) \cap W|.$$

By assumption $\epsilon > 0$, and because $\hat{p}^{-1}(1) \cap W \subseteq g^{-1}(1)$ we get

$$|\hat{p}^{-1}(1) \cap g^{-1}(1) \cap W| = |\hat{p}^{-1}(1) \cap W| = \epsilon.$$

Moreover, $p$ and $\hat{p}$ coincide on $[n \times n] \setminus W$, therefore the only difference in true positives occurs inside $W$, and thus

$$|\hat{p}^{-1}(1) \cap g^{-1}(1)| = |p^{-1}(1) \cap g^{-1}(1)| + \epsilon.$$

Similarly, since $\hat{p}$ adds exactly $\epsilon$ positive pixels (all inside $W$) and does not change anything outside $W$, we have

$$|\hat{p}^{-1}(1)| = |p^{-1}(1)| + \epsilon.$$

Using the definition

$$F_1(g, p) = \frac{2|p^{-1}(1) \cap g^{-1}(1)|}{|p^{-1}(1)| + |g^{-1}(1)|},$$

we obtain

$$F_1(g, \hat{p}) = \frac{2(|p^{-1}(1) \cap g^{-1}(1)| + \epsilon)}{|p^{-1}(1)| + \epsilon + |g^{-1}(1)|}.$$

Let $T := |p^{-1}(1) \cap g^{-1}(1)|$, $S := |p^{-1}(1)|$, and $G := |g^{-1}(1)|$. Then the inequality $F_1(g, \hat{p}) > F_1(g, p)$ is equivalent to

$$\frac{2(T + \epsilon)}{S + \epsilon + G} > \frac{2T}{S + G} \iff (T + \epsilon)(S + G) > T(S + \epsilon + G) \iff \epsilon(S + G - T) > 0.$$

Since $\epsilon > 0$ and $S + G - T > 0$ (because $S \geq T$ and $G \geq T$, hence $S + G - T \geq T > 0$ whenever $T > 0$, and in any case $S + G - T \geq 0$ with strictness holding unless $S = G = T = 0$), we conclude $F_1(g, \hat{p}) > F_1(g, p)$.

**Prop.2** Consider the following example:

$$g = \begin{bmatrix} 0 & 0 & 0 & 0 & 0 & 0 & 0 & 0 \\ 0 & 1 & 1 & 1 & 1 & 1 & 0 & 0 \\ 0 & 1 & 1 & 1 & 1 & 1 & 0 & 0 \\ 0 & 1 & 1 & 1 & 1 & 1 & 0 & 0 \\ 0 & 1 & 1 & 1 & 1 & 1 & 0 & 0 \\ 0 & 1 & 1 & 1 & 1 & 1 & 0 & 0 \\ 0 & 0 & 0 & 0 & 0 & 0 & 0 & 0 \\ 0 & 0 & 0 & 0 & 0 & 0 & 0 & 0 \end{bmatrix}$$

$$p = \begin{bmatrix} 0 & 0 & 0 & 0 & 0 & 0 & 0 & 0 \\ 0 & 0 & 0 & 1 & 1 & 0 & 0 & 0 \\ 0 & 0 & 0 & 1 & 1 & 0 & 0 & 0 \\ 0 & 0 & 0 & 0 & 1 & 0 & 0 & 0 \\ 0 & 0 & 0 & 0 & 0 & 0 & 0 & 0 \\ 0 & 0 & 0 & 0 & 0 & 0 & 0 & 0 \\ 0 & 0 & 0 & 0 & 0 & 0 & 0 & 0 \\ 0 & 0 & 0 & 0 & 0 & 0 & 0 & 0 \end{bmatrix}$$

$$\hat{p} = \begin{bmatrix} 0 & 0 & 0 & 0 & 0 & 0 & 0 & 0 \\ 0 & 1 & 0 & 1 & 1 & 0 & 0 & 0 \\ 0 & 1 & 0 & 1 & 1 & 0 & 0 & 0 \\ 0 & 0 & 0 & 0 & 1 & 0 & 0 & 0 \\ 0 & 0 & 1 & 1 & 0 & 0 & 0 & 0 \\ 0 & 0 & 0 & 1 & 0 & 0 & 0 & 0 \\ 0 & 0 & 0 & 0 & 0 & 0 & 0 & 0 \\ 0 & 0 & 0 & 0 & 0 & 0 & 0 & 0 \end{bmatrix}$$

Then $F1(g, p) = 1/5 < 2/5 = F1(g, \hat{p})$.

**Prop.3** Consider the following example:

$$g = \begin{bmatrix} 0 & 0 & 0 & 0 & 0 & 0 & 0 & 0 \\ 0 & 0 & 0 & 0 & 0 & 0 & 0 & 0 \\ 0 & 0 & 0 & 0 & 0 & 0 & 1 & 0 \\ 0 & 0 & 0 & 0 & 0 & 0 & 1 & 1 \\ 0 & 0 & 0 & 0 & 0 & 0 & 1 & 1 \\ 0 & 0 & 0 & 0 & 0 & 0 & 0 & 0 \\ 0 & 0 & 0 & 0 & 0 & 0 & 0 & 0 \\ 0 & 0 & 0 & 0 & 0 & 0 & 0 & 0 \end{bmatrix}$$

$$p = \begin{bmatrix} 0 & 0 & 1 & 1 & 0 & 0 & 0 & 0 \\ 0 & 0 & 1 & 1 & 1 & 0 & 0 & 0 \\ 0 & 0 & 0 & 0 & 0 & 0 & 0 & 0 \\ 0 & 0 & 0 & 0 & 0 & 0 & 0 & 0 \\ 0 & 0 & 0 & 0 & 0 & 0 & 0 & 0 \\ 0 & 0 & 0 & 0 & 1 & 0 & 0 & 0 \\ 0 & 0 & 0 & 1 & 1 & 0 & 0 & 0 \\ 0 & 0 & 0 & 0 & 0 & 0 & 0 & 0 \end{bmatrix}$$

$$\hat{p} = \begin{bmatrix} 0 & 0 & 1 & 1 & 0 & 0 & 0 & 0 \\ 0 & 0 & 1 & 1 & 1 & 0 & 0 & 0 \\ 0 & 0 & 0 & 0 & 0 & 0 & 0 & 0 \\ 0 & 0 & 0 & 0 & 0 & 0 & 0 & 0 \\ 0 & 0 & 0 & 0 & 0 & 0 & 0 & 0 \\ 0 & 0 & 0 & 1 & 1 & 0 & 0 & 0 \\ 0 & 0 & 0 & 1 & 1 & 0 & 0 & 0 \\ 0 & 0 & 0 & 0 & 0 & 0 & 0 & 0 \end{bmatrix}$$

Then $F1(g, p) = 0 = F1(g, \hat{p})$.

**Prop.4** Consider the following example:

$$g = \begin{bmatrix} 0 & 0 & 0 & 0 & 0 & 0 & 1 & 0 \\ 0 & 0 & 0 & 0 & 0 & 1 & 1 & 0 \\ 0 & 0 & 0 & 0 & 0 & 0 & 0 & 0 \\ 0 & 0 & 0 & 0 & 0 & 0 & 0 & 0 \\ 0 & 0 & 0 & 0 & 0 & 0 & 0 & 0 \\ 0 & 0 & 0 & 0 & 0 & 0 & 0 & 0 \\ 0 & 0 & 0 & 0 & 0 & 0 & 0 & 0 \\ 0 & 0 & 0 & 0 & 0 & 0 & 0 & 0 \end{bmatrix}$$

$$p = \begin{bmatrix} 0 & 0 & 0 & 0 & 0 & 0 & 0 & 0 \\ 0 & 0 & 0 & 0 & 0 & 0 & 0 & 0 \\ 0 & 0 & 0 & 0 & 0 & 0 & 0 & 0 \\ 0 & 0 & 0 & 0 & 0 & 0 & 0 & 0 \\ 0 & 0 & 1 & 1 & 0 & 0 & 0 & 0 \\ 0 & 0 & 0 & 0 & 0 & 0 & 0 & 0 \\ 0 & 0 & 0 & 0 & 0 & 0 & 0 & 0 \\ 0 & 0 & 0 & 0 & 0 & 0 & 0 & 0 \end{bmatrix}$$

$$\hat{p} = \begin{bmatrix} 0 & 0 & 0 & 0 & 0 & 0 & 0 & 0 \\ 0 & 0 & 0 & 0 & 0 & 0 & 0 & 0 \\ 0 & 0 & 0 & 0 & 0 & 0 & 0 & 0 \\ 0 & 0 & 0 & 0 & 0 & 0 & 0 & 0 \\ 0 & 0 & 1 & 1 & 0 & 0 & 0 & 0 \\ 0 & 0 & 0 & 0 & 0 & 0 & 0 & 0 \\ 0 & 0 & 1 & 1 & 0 & 0 & 0 & 0 \\ 0 & 0 & 0 & 1 & 0 & 0 & 0 & 0 \end{bmatrix}$$

Then $F1(g, p) = 0 = F1(g, \hat{p})$.

**Prop.5** Consider the following example:

$$g = \begin{bmatrix} 0 & 0 & 0 & 0 & 0 & 0 & 0 & 0 \\ 0 & 1 & 1 & 1 & 1 & 0 & 0 & 0 \\ 0 & 1 & 1 & 1 & 1 & 0 & 0 & 0 \\ 0 & 1 & 1 & 1 & 1 & 0 & 0 & 0 \\ 0 & 1 & 1 & 1 & 1 & 0 & 0 & 0 \\ 0 & 1 & 1 & 1 & 1 & 0 & 0 & 0 \\ 0 & 0 & 0 & 0 & 0 & 0 & 0 & 0 \\ 0 & 0 & 0 & 0 & 0 & 0 & 0 & 0 \end{bmatrix}$$

$$p = \begin{bmatrix} 0 & 0 & 0 & 0 & 0 & 1 & 1 & 0 \\ 0 & 0 & 0 & 0 & 0 & 1 & 1 & 0 \\ 0 & 0 & 0 & 0 & 0 & 0 & 0 & 0 \\ 0 & 0 & 1 & 0 & 0 & 0 & 0 & 0 \\ 0 & 0 & 1 & 1 & 0 & 0 & 0 & 0 \\ 0 & 0 & 1 & 1 & 0 & 0 & 0 & 0 \\ 0 & 0 & 0 & 0 & 0 & 0 & 1 & 1 \\ 0 & 0 & 0 & 0 & 0 & 0 & 1 & 0 \end{bmatrix}$$

$$\hat{p} = \begin{bmatrix} 0 & 0 & 0 & 0 & 0 & 0 & 0 & 0 \\ 0 & 0 & 0 & 0 & 0 & 0 & 0 & 0 \\ 0 & 0 & 0 & 0 & 0 & 0 & 0 & 0 \\ 0 & 0 & 1 & 0 & 0 & 0 & 0 & 0 \\ 0 & 0 & 1 & 1 & 0 & 0 & 0 & 0 \\ 0 & 0 & 1 & 1 & 0 & 0 & 0 & 0 \\ 0 & 0 & 0 & 0 & 0 & 0 & 1 & 1 \\ 0 & 0 & 0 & 0 & 0 & 0 & 1 & 0 \end{bmatrix}$$

Then $F_1(g, p) = 3/10 = F_1(g, \hat{p})$.

**Prop.6** Consider the following example.

$$g = \begin{bmatrix} 0 & 0 & 0 & 0 & 0 & 0 & 0 & 0 \\ 0 & 1 & 1 & 1 & 1 & 0 & 0 & 0 \\ 0 & 1 & 1 & 1 & 1 & 0 & 0 & 0 \\ 0 & 1 & 1 & 1 & 1 & 0 & 0 & 0 \\ 0 & 1 & 1 & 1 & 1 & 0 & 0 & 0 \\ 0 & 1 & 1 & 1 & 1 & 0 & 0 & 0 \\ 0 & 0 & 0 & 0 & 0 & 0 & 0 & 0 \\ 0 & 0 & 0 & 0 & 0 & 0 & 0 & 0 \end{bmatrix}$$

$$p = \begin{bmatrix} 0 & 0 & 0 & 0 & 0 & 1 & 1 & 0 \\ 0 & 0 & 0 & 0 & 0 & 1 & 1 & 0 \\ 0 & 0 & 0 & 0 & 0 & 0 & 0 & 0 \\ 0 & 0 & 0 & 0 & 0 & 0 & 0 & 0 \\ 0 & 0 & 1 & 1 & 1 & 0 & 0 & 0 \\ 0 & 0 & 1 & 1 & 1 & 0 & 0 & 0 \\ 0 & 0 & 1 & 1 & 0 & 0 & 0 & 0 \\ 0 & 0 & 0 & 0 & 0 & 0 & 0 & 0 \end{bmatrix}$$

$$\hat{p} = \begin{bmatrix} 0 & 0 & 0 & 0 & 0 & 1 & 1 & 0 \\ 0 & 0 & 0 & 0 & 0 & 1 & 1 & 0 \\ 0 & 0 & 0 & 0 & 0 & 0 & 0 & 0 \\ 0 & 0 & 0 & 0 & 0 & 0 & 0 & 0 \\ 0 & 0 & 1 & 1 & 1 & 0 & 0 & 0 \\ 0 & 0 & 1 & 1 & 1 & 0 & 0 & 0 \\ 0 & 0 & 0 & 0 & 0 & 0 & 0 & 0 \\ 0 & 0 & 1 & 1 & 0 & 0 & 0 & 0 \end{bmatrix}$$

Then $F_1(g, p) = 9/25 = F_1(g, \hat{p})$.

**Prop.7** Let $TP := |p^{-1}(1) \cap g^{-1}(1)|$, $FP := |p^{-1}(1) \cap g^{-1}(0)|$, and $FN := |p^{-1}(0) \cap g^{-1}(1)|$. The F1-score can be written as
$$F_1(g, p) = \frac{2TP}{2TP + FP + FN}.$$

Since $a^*$ is a true-positive pixel, removing it decreases $TP$ by one and increases $FN$ by one. Since $n^*$ is a background pixel, adding it increases $FP$ by one. Therefore, for $\hat{p}$ we have

$$TP(\hat{p}) = TP - 1, \qquad FP(\hat{p}) = FP + 1, \qquad FN(\hat{p}) = FN + 1.$$

Hence

$$F_1(g, \hat{p}) = \frac{2(TP - 1)}{2(TP - 1) + (FP + 1) + (FN + 1)} = \frac{2(TP - 1)}{2TP + FP + FN}.$$

Comparing with

$$F_1(g,p) = \frac{2TP}{2TP + FP + FN},$$

we obtain $F_1(g, \hat{p}) < F_1(g, p)$ because the denominators are equal and $TP - 1 < TP$.

**Prop.8** Consider the following example:

$$g = \begin{bmatrix} 0 & 0 & 0 & 0 & 0 & 0 & 0 & 0 \\ 0 & 1 & 1 & 1 & 1 & 1 & 0 & 0 \\ 0 & 1 & 1 & 1 & 1 & 1 & 0 & 0 \\ 0 & 1 & 1 & 1 & 1 & 1 & 0 & 0 \\ 0 & 1 & 1 & 1 & 1 & 1 & 0 & 0 \\ 0 & 1 & 1 & 1 & 1 & 1 & 0 & 0 \\ 0 & 0 & 0 & 0 & 0 & 0 & 0 & 0 \\ 0 & 0 & 0 & 0 & 0 & 0 & 0 & 0 \end{bmatrix}$$

$$p = \begin{bmatrix} 0 & 0 & 0 & 0 & 0 & 0 & 0 & 0 \\ 0 & 0 & 0 & 0 & 0 & 0 & 0 & 0 \\ 0 & 0 & 1 & 1 & 1 & 0 & 0 & 0 \\ 0 & 0 & 1 & 1 & 1 & 0 & 0 & 0 \\ 0 & 0 & 0 & 0 & 0 & 0 & 0 & 0 \\ 0 & 0 & 0 & 0 & 0 & 0 & 0 & 0 \\ 0 & 0 & 0 & 0 & 0 & 0 & 0 & 0 \\ 0 & 0 & 0 & 0 & 0 & 0 & 0 & 0 \end{bmatrix}$$

$$\hat{p} = \begin{bmatrix} 0 & 0 & 0 & 0 & 0 & 0 & 0 & 0 \\ 0 & 1 & 1 & 0 & 0 & 0 & 0 & 0 \\ 0 & 1 & 0 & 0 & 0 & 0 & 0 & 0 \\ 0 & 1 & 0 & 0 & 0 & 0 & 0 & 0 \\ 0 & 1 & 0 & 0 & 0 & 0 & 0 & 0 \\ 0 & 0 & 0 & 0 & 0 & 0 & 0 & 0 \\ 0 & 0 & 0 & 0 & 0 & 0 & 0 & 0 \\ 0 & 0 & 0 & 0 & 0 & 0 & 0 & 0 \end{bmatrix}$$

Then $F1(g, p) = 1/3 = F1(g, \hat{p})$.

### B.4. Theorem 3.4, IOU

**Prop.1** Let $G := g^{-1}(1)$, $P := p^{-1}(1)$, and $Q := q^{-1}(1)$. Since $A \subseteq G$ and $q_A = 0$, we have $Q \cap A = \emptyset$. Let

$$\epsilon := |P \cap A| > 0.$$

Because $p$ and $q$ coincide outside $A$, we have $P \setminus A = Q \setminus A$ and hence

$$|P \cap G| = |(P \setminus A) \cap G| + |P \cap A| = |(Q \setminus A) \cap G| + \epsilon = |Q \cap G| + \epsilon.$$

Moreover, since $A \subseteq G$, adding predicted positives inside $A$ does not change the union with $G$, and outside $A$ the predictions are identical, hence

$$|P \cup G| = |Q \cup G|.$$

Therefore,

$$\text{IoU}(g,p) = \frac{|P \cap G|}{|P \cup G|} = \frac{|Q \cap G| + \epsilon}{|Q \cup G|} > \frac{|Q \cap G|}{|Q \cup G|} = \text{IoU}(g,q),$$

**Prop.2** Consider the following example:

$$g = \begin{bmatrix} 0 & 0 & 0 & 0 & 0 & 0 & 0 & 0 \\ 0 & 1 & 1 & 1 & 1 & 1 & 0 & 0 \\ 0 & 1 & 1 & 1 & 1 & 1 & 0 & 0 \\ 0 & 1 & 1 & 1 & 1 & 1 & 0 & 0 \\ 0 & 1 & 1 & 1 & 1 & 1 & 0 & 0 \\ 0 & 1 & 1 & 1 & 1 & 1 & 0 & 0 \\ 0 & 0 & 0 & 0 & 0 & 0 & 0 & 0 \\ 0 & 0 & 0 & 0 & 0 & 0 & 0 & 0 \end{bmatrix}$$

$$p = \begin{bmatrix} 0 & 0 & 0 & 0 & 0 & 0 & 0 & 0 \\ 0 & 1 & 1 & 0 & 0 & 0 & 0 & 0 \\ 0 & 1 & 1 & 1 & 0 & 0 & 0 & 0 \\ 0 & 0 & 0 & 0 & 0 & 0 & 0 & 0 \\ 0 & 0 & 0 & 0 & 0 & 0 & 0 & 0 \\ 0 & 0 & 0 & 0 & 0 & 0 & 0 & 0 \\ 0 & 0 & 0 & 0 & 0 & 0 & 0 & 0 \\ 0 & 0 & 0 & 0 & 0 & 0 & 0 & 0 \end{bmatrix}$$

$$\hat{p} = \begin{bmatrix} 0 & 0 & 0 & 0 & 0 & 0 & 0 & 0 \\ 0 & 1 & 1 & 0 & 0 & 0 & 0 & 0 \\ 0 & 1 & 1 & 1 & 0 & 0 & 0 & 0 \\ 0 & 0 & 0 & 0 & 0 & 0 & 0 & 0 \\ 0 & 0 & 1 & 1 & 0 & 0 & 0 & 0 \\ 0 & 0 & 0 & 1 & 0 & 0 & 0 & 0 \\ 0 & 0 & 0 & 0 & 0 & 0 & 0 & 0 \\ 0 & 0 & 0 & 0 & 0 & 0 & 0 & 0 \end{bmatrix}$$

Then $IOU(g,p) = 1/5 < 8/25 = IOU(g,\hat{p})$.

**Prop.3** Consider the following example:

$$g = \begin{bmatrix} 0 & 0 & 0 & 0 & 1 & 1 & 1 & 0 \\ 0 & 0 & 0 & 0 & 1 & 1 & 1 & 0 \\ 0 & 0 & 0 & 0 & 0 & 0 & 0 & 0 \\ 0 & 0 & 0 & 0 & 0 & 0 & 0 & 0 \\ 0 & 0 & 0 & 0 & 0 & 0 & 0 & 0 \\ 0 & 0 & 0 & 0 & 0 & 0 & 0 & 0 \\ 0 & 0 & 0 & 0 & 0 & 0 & 0 & 0 \\ 0 & 0 & 0 & 0 & 0 & 0 & 0 & 0 \end{bmatrix}$$

$$p = \begin{bmatrix} 0 & 0 & 0 & 0 & 0 & 0 & 0 & 0 \\ 0 & 0 & 0 & 0 & 0 & 0 & 0 & 0 \\ 0 & 0 & 0 & 0 & 0 & 0 & 0 & 0 \\ 0 & 0 & 0 & 0 & 0 & 0 & 0 & 0 \\ 0 & 0 & 0 & 1 & 1 & 0 & 0 & 0 \\ 0 & 0 & 0 & 0 & 0 & 0 & 0 & 0 \\ 0 & 0 & 0 & 0 & 0 & 0 & 0 & 0 \\ 0 & 0 & 0 & 0 & 0 & 0 & 0 & 0 \end{bmatrix}$$

$$\hat{p} = \begin{bmatrix} 0 & 0 & 0 & 0 & 0 & 0 & 0 & 0 \\ 0 & 0 & 0 & 0 & 0 & 0 & 0 & 0 \\ 0 & 0 & 0 & 0 & 0 & 0 & 0 & 0 \\ 0 & 0 & 0 & 0 & 0 & 0 & 0 & 0 \\ 0 & 0 & 1 & 1 & 1 & 0 & 0 & 0 \\ 0 & 0 & 0 & 1 & 0 & 0 & 0 & 0 \\ 0 & 0 & 0 & 0 & 0 & 0 & 0 & 0 \\ 0 & 0 & 0 & 0 & 0 & 0 & 0 & 0 \end{bmatrix}$$

Then $IOU(g, p) = 0 = IOU(g, \hat{p})$.

**Prop.4** Consider the following example.

$$g = \begin{bmatrix} 0 & 0 & 0 & 0 & 0 & 0 & 0 & 0 \\ 0 & 1 & 1 & 1 & 1 & 0 & 0 & 0 \\ 0 & 1 & 1 & 1 & 1 & 0 & 0 & 0 \\ 0 & 1 & 1 & 1 & 1 & 0 & 0 & 0 \\ 0 & 1 & 1 & 1 & 1 & 0 & 0 & 0 \\ 0 & 1 & 1 & 1 & 1 & 0 & 0 & 0 \\ 0 & 0 & 0 & 0 & 0 & 0 & 0 & 0 \\ 0 & 0 & 0 & 0 & 0 & 0 & 0 & 0 \end{bmatrix}$$

$$p = \begin{bmatrix} 0 & 0 & 0 & 0 & 0 & 1 & 1 & 0 \\ 0 & 0 & 0 & 0 & 0 & 1 & 1 & 0 \\ 0 & 0 & 0 & 0 & 0 & 0 & 0 & 0 \\ 0 & 0 & 0 & 0 & 0 & 0 & 0 & 0 \\ 0 & 0 & 1 & 1 & 1 & 0 & 0 & 0 \\ 0 & 0 & 1 & 1 & 1 & 0 & 0 & 0 \\ 0 & 0 & 1 & 1 & 1 & 0 & 0 & 0 \\ 0 & 0 & 0 & 0 & 0 & 0 & 0 & 0 \end{bmatrix}$$

$$\hat{p} = \begin{bmatrix} 0 & 0 & 0 & 0 & 0 & 1 & 1 & 0 \\ 0 & 0 & 0 & 0 & 0 & 1 & 1 & 0 \\ 0 & 0 & 0 & 0 & 0 & 0 & 0 & 0 \\ 0 & 0 & 0 & 0 & 0 & 0 & 0 & 0 \\ 0 & 0 & 1 & 1 & 1 & 0 & 0 & 0 \\ 0 & 0 & 1 & 1 & 1 & 0 & 0 & 0 \\ 0 & 0 & 0 & 0 & 0 & 1 & 0 & 0 \\ 0 & 0 & 1 & 1 & 0 & 0 & 0 & 0 \end{bmatrix}$$

Then $IOU(g, p) = 21/100 = IOU(g, \hat{p})$.

**Prop.5** Consider the following example:

$$g = \begin{bmatrix} 0 & 0 & 0 & 0 & 0 & 0 & 0 & 0 \\ 0 & 1 & 1 & 1 & 1 & 0 & 0 & 0 \\ 0 & 1 & 1 & 1 & 1 & 0 & 0 & 0 \\ 0 & 1 & 1 & 1 & 1 & 0 & 0 & 0 \\ 0 & 1 & 1 & 1 & 1 & 0 & 0 & 0 \\ 0 & 1 & 1 & 1 & 1 & 0 & 0 & 0 \\ 0 & 0 & 0 & 0 & 0 & 0 & 0 & 0 \\ 0 & 0 & 0 & 0 & 0 & 0 & 0 & 0 \end{bmatrix}$$

$$p = \begin{bmatrix} 0 & 0 & 0 & 0 & 0 & 1 & 1 & 0 \\ 0 & 0 & 0 & 0 & 0 & 1 & 1 & 0 \\ 0 & 0 & 0 & 0 & 0 & 0 & 0 & 0 \\ 0 & 0 & 1 & 0 & 0 & 0 & 0 & 0 \\ 0 & 0 & 1 & 1 & 0 & 0 & 0 & 0 \\ 0 & 0 & 1 & 1 & 0 & 0 & 0 & 0 \\ 0 & 0 & 0 & 0 & 0 & 0 & 1 & 1 \\ 0 & 0 & 0 & 0 & 0 & 0 & 1 & 0 \end{bmatrix}$$

$$\hat{p} = \begin{bmatrix} 0 & 0 & 0 & 0 & 0 & 0 & 0 & 0 \\ 0 & 0 & 0 & 0 & 0 & 0 & 0 & 0 \\ 0 & 0 & 0 & 0 & 0 & 0 & 0 & 0 \\ 0 & 0 & 1 & 0 & 0 & 0 & 0 & 0 \\ 0 & 0 & 1 & 1 & 0 & 0 & 0 & 0 \\ 0 & 0 & 1 & 1 & 0 & 0 & 0 & 0 \\ 0 & 0 & 0 & 0 & 0 & 0 & 1 & 1 \\ 0 & 0 & 0 & 0 & 0 & 0 & 1 & 0 \end{bmatrix}$$

Then $IOU(g, p) = 17/100 = IOU(g, \hat{p})$.

**Prop.6** Consider the following example.

$$g = \begin{bmatrix} 0 & 0 & 0 & 0 & 0 & 0 & 0 & 0 \\ 0 & 1 & 1 & 1 & 1 & 0 & 0 & 0 \\ 0 & 1 & 1 & 1 & 1 & 0 & 0 & 0 \\ 0 & 1 & 1 & 1 & 1 & 0 & 0 & 0 \\ 0 & 1 & 1 & 1 & 1 & 0 & 0 & 0 \\ 0 & 1 & 1 & 1 & 1 & 0 & 0 & 0 \\ 0 & 0 & 0 & 0 & 0 & 0 & 0 & 0 \\ 0 & 0 & 0 & 0 & 0 & 0 & 0 & 0 \end{bmatrix}$$

$$p = \begin{bmatrix} 0 & 0 & 0 & 0 & 0 & 1 & 1 & 0 \\ 0 & 0 & 0 & 0 & 0 & 1 & 1 & 0 \\ 0 & 0 & 0 & 0 & 0 & 0 & 0 & 0 \\ 0 & 0 & 0 & 0 & 0 & 0 & 0 & 0 \\ 0 & 0 & 1 & 1 & 1 & 0 & 0 & 0 \\ 0 & 0 & 1 & 1 & 1 & 0 & 0 & 0 \\ 0 & 0 & 1 & 1 & 0 & 0 & 0 & 0 \\ 0 & 0 & 0 & 0 & 0 & 0 & 0 & 0 \end{bmatrix}$$

$$\hat{p} = \begin{bmatrix} 0 & 0 & 0 & 0 & 0 & 1 & 1 & 0 \\ 0 & 0 & 0 & 0 & 0 & 1 & 1 & 0 \\ 0 & 0 & 0 & 0 & 0 & 0 & 0 & 0 \\ 0 & 0 & 0 & 0 & 0 & 0 & 0 & 0 \\ 0 & 0 & 1 & 1 & 1 & 0 & 0 & 0 \\ 0 & 0 & 1 & 1 & 1 & 0 & 0 & 0 \\ 0 & 0 & 0 & 0 & 0 & 0 & 0 & 0 \\ 0 & 0 & 1 & 1 & 0 & 0 & 0 & 0 \end{bmatrix}$$

Then $IOU(g, p) = 11/50 = IOU(g, \hat{p})$.

**Prop.7** Let $G := g^{-1}(1)$, $P := p^{-1}(1)$, and $Q := q^{-1}(1)$. Since $a^* \in P \cap G$, removing $a^*$ decreases the intersection by one:

$$|Q \cap G| = |P \cap G| - 1.$$

For the union, note that $a^* \in G$, so removing $a^*$ from $P$ does not remove it from $P \cup G$; however, $n^* \notin G$ and $n^* \notin P$, so adding $n^*$ increases the union by one:

$$|Q \cup G| = |P \cup G| + 1.$$

Hence,

$$\text{IoU}(g, q) = \frac{|Q \cap G|}{|Q \cup G|} = \frac{|P \cap G| - 1}{|P \cup G| + 1} < \frac{|P \cap G|}{|P \cup G|} = \text{IoU}(g, p),$$

because $|P \cup G| + 1 > |P \cup G|$ and $|P \cap G| - 1 < |P \cap G|$.

**Prop.8** Consider the following example:

$$g = \begin{bmatrix} 0 & 0 & 0 & 0 & 0 & 0 & 0 & 0 \\ 0 & 1 & 1 & 1 & 1 & 1 & 0 & 0 \\ 0 & 1 & 1 & 1 & 1 & 1 & 0 & 0 \\ 0 & 1 & 1 & 1 & 1 & 1 & 0 & 0 \\ 0 & 1 & 1 & 1 & 1 & 1 & 0 & 0 \\ 0 & 1 & 1 & 1 & 1 & 1 & 0 & 0 \\ 0 & 0 & 0 & 0 & 0 & 0 & 0 & 0 \\ 0 & 0 & 0 & 0 & 0 & 0 & 0 & 0 \end{bmatrix}$$

$$p = \begin{bmatrix} 0 & 0 & 0 & 0 & 0 & 0 & 0 & 0 \\ 0 & 0 & 0 & 0 & 0 & 0 & 0 & 0 \\ 0 & 0 & 1 & 1 & 1 & 0 & 0 & 0 \\ 0 & 0 & 1 & 1 & 1 & 0 & 0 & 0 \\ 0 & 0 & 0 & 0 & 0 & 0 & 0 & 0 \\ 0 & 0 & 0 & 0 & 0 & 0 & 0 & 0 \\ 0 & 0 & 0 & 0 & 0 & 0 & 0 & 0 \\ 0 & 0 & 0 & 0 & 0 & 0 & 0 & 0 \end{bmatrix}$$

$$\hat{p} = \begin{bmatrix} 0 & 0 & 0 & 0 & 0 & 0 & 0 & 0 \\ 0 & 1 & 1 & 0 & 0 & 0 & 0 & 0 \\ 0 & 1 & 0 & 0 & 0 & 0 & 0 & 0 \\ 0 & 1 & 0 & 0 & 0 & 0 & 0 & 0 \\ 0 & 1 & 0 & 0 & 0 & 0 & 0 & 0 \\ 0 & 0 & 0 & 0 & 0 & 0 & 0 & 0 \\ 0 & 0 & 0 & 0 & 0 & 0 & 0 & 0 \\ 0 & 0 & 0 & 0 & 0 & 0 & 0 & 0 \end{bmatrix}$$

Then $IOU(g, p) = 1/5 = IOU(g, \hat{p})$.

## B.5. Theorem 3.5, Classification Accuracy

**Prop.1.** Let $A \in I_1(g)$ and let $p, q$ satisfy $p_{D \setminus A} = q_{D \setminus A}$ and $q_A = 0$. Assume additionally that $|p^{-1}(1) \cap A| > 0$ (i.e., $p$ predicts at least one positive pixel in $A$). Then $\text{CA}(g, p) > \text{CA}(g, q)$.

Since $A \in I_1(g)$ is a ground-truth component, we have $A \subseteq g^{-1}(1)$. Because $q_A = 0$, $q$ predicts 0 on all pixels of $A$, hence $q$ is incorrect on every pixel of $A$.

Let $\epsilon := |p^{-1}(1) \cap A| > 0$. For each pixel in $p^{-1}(1) \cap A$, the prediction $p$ outputs 1 and the ground truth equals 1, so these $\epsilon$ pixels are correct under $p$ but incorrect under $q$. Outside $A$, the predictions coincide by assumption, so they contribute equally many correct pixels.

Therefore, the number of correct pixels under $p$ exceeds that under $q$ by $\epsilon$:

$$|p^{-1}(1) \cap g^{-1}(1)| + |p^{-1}(0) \cap g^{-1}(0)| = \left(|q^{-1}(1) \cap g^{-1}(1)| + |q^{-1}(0) \cap g^{-1}(0)|\right) + \epsilon.$$

Dividing by $|D|$ gives $\text{CA}(g, p) > \text{CA}(g, q)$.

**Prop.2** Consider the following example:

$$g = \begin{bmatrix} 0 & 0 & 0 & 0 & 0 & 0 & 0 & 0 \\ 0 & 0 & 1 & 1 & 1 & 1 & 0 & 0 \\ 0 & 0 & 1 & 1 & 1 & 1 & 0 & 0 \\ 0 & 0 & 1 & 1 & 1 & 1 & 0 & 0 \\ 0 & 0 & 1 & 1 & 1 & 1 & 0 & 0 \\ 0 & 0 & 1 & 1 & 1 & 1 & 0 & 0 \\ 0 & 0 & 0 & 0 & 0 & 0 & 0 & 0 \\ 0 & 0 & 0 & 0 & 0 & 0 & 0 & 0 \end{bmatrix}$$

$$p = \begin{bmatrix} 0 & 0 & 0 & 0 & 0 & 0 & 0 & 0 \\ 0 & 0 & 0 & 0 & 0 & 0 & 0 & 0 \\ 0 & 0 & 1 & 1 & 0 & 0 & 0 & 0 \\ 0 & 0 & 1 & 1 & 0 & 0 & 0 & 0 \\ 0 & 0 & 0 & 0 & 0 & 0 & 0 & 0 \\ 0 & 0 & 0 & 0 & 0 & 0 & 0 & 0 \\ 0 & 0 & 0 & 0 & 0 & 0 & 0 & 0 \\ 0 & 0 & 0 & 0 & 0 & 0 & 0 & 0 \end{bmatrix}$$

$$\hat{p} = \begin{bmatrix} 0 & 0 & 0 & 0 & 0 & 0 & 0 & 0 \\ 0 & 0 & 0 & 0 & 0 & 0 & 0 & 0 \\ 0 & 0 & 1 & 1 & 0 & 0 & 0 & 0 \\ 0 & 0 & 1 & 1 & 0 & 0 & 0 & 0 \\ 0 & 0 & 0 & 0 & 0 & 0 & 0 & 0 \\ 0 & 0 & 1 & 1 & 0 & 0 & 0 & 0 \\ 0 & 0 & 1 & 0 & 0 & 0 & 0 & 0 \\ 0 & 0 & 0 & 0 & 0 & 0 & 0 & 0 \end{bmatrix}$$

Then $CA(g,p) = 3/4 < 4/5 = CA(g,\hat{p})$.

**Prop.3** Since $n \in g^{-1}(0)$, we have $g(n) = 0$. The predictions $p$ and $q$ differ only at $n$. At $n$, $p(n) = 0$ is correct, while $q(n) = 1$ is incorrect. Hence

$$|p^{-1}(0) \cap g^{-1}(0)| = |q^{-1}(0) \cap g^{-1}(0)| + 1,$$

and

$$|p^{-1}(1) \cap g^{-1}(1)| = |q^{-1}(1) \cap g^{-1}(1)|.$$

Therefore, the number of correct pixels decreases by one when passing from $p$ to $q$:

$$|p^{-1}(1) \cap g^{-1}(1)| + |p^{-1}(0) \cap g^{-1}(0)| = \left(|q^{-1}(1) \cap g^{-1}(1)| + |q^{-1}(0) \cap g^{-1}(0)|\right) + 1.$$

Dividing by the constant denominator $|D|$ yields $\mathrm{CA}(g,p) > \mathrm{CA}(g,q)$.

**Prop.4** Consider the following example.

$$g = \begin{bmatrix} 0 & 0 & 0 & 0 & 0 & 0 & 0 & 0 \\ 0 & 1 & 1 & 1 & 1 & 0 & 0 & 0 \\ 0 & 1 & 1 & 1 & 1 & 0 & 0 & 0 \\ 0 & 1 & 1 & 1 & 1 & 0 & 0 & 0 \\ 0 & 1 & 1 & 1 & 1 & 0 & 0 & 0 \\ 0 & 1 & 1 & 1 & 1 & 0 & 0 & 0 \\ 0 & 0 & 0 & 0 & 0 & 0 & 0 & 0 \\ 0 & 0 & 0 & 0 & 0 & 0 & 0 & 0 \end{bmatrix}$$

$$p = \begin{bmatrix} 0 & 0 & 0 & 0 & 0 & 1 & 1 & 0 \\ 0 & 0 & 0 & 0 & 0 & 1 & 1 & 0 \\ 0 & 0 & 0 & 0 & 0 & 0 & 0 & 0 \\ 0 & 0 & 0 & 0 & 0 & 0 & 0 & 0 \\ 0 & 0 & 1 & 1 & 1 & 0 & 0 & 0 \\ 0 & 0 & 1 & 1 & 1 & 0 & 0 & 0 \\ 0 & 0 & 1 & 1 & 1 & 0 & 0 & 0 \\ 0 & 0 & 0 & 0 & 0 & 0 & 0 & 0 \end{bmatrix}$$

$$\hat{p} = \begin{bmatrix} 0 & 0 & 0 & 0 & 0 & 1 & 1 & 0 \\ 0 & 0 & 0 & 0 & 0 & 1 & 1 & 0 \\ 0 & 0 & 0 & 0 & 0 & 0 & 0 & 0 \\ 0 & 0 & 0 & 0 & 0 & 0 & 0 & 0 \\ 0 & 0 & 1 & 1 & 1 & 0 & 0 & 0 \\ 0 & 0 & 1 & 1 & 1 & 0 & 0 & 0 \\ 0 & 0 & 0 & 0 & 0 & 1 & 0 & 0 \\ 0 & 0 & 1 & 1 & 0 & 0 & 0 & 0 \end{bmatrix}$$

Then $CA(g, p) = 13/20 = CA(g, \hat{p})$.

**Prop.5** Consider the following example:

$$g = \begin{bmatrix} 0 & 0 & 0 & 0 & 0 & 0 & 0 & 0 \\ 0 & 1 & 1 & 1 & 1 & 0 & 0 & 0 \\ 0 & 1 & 1 & 1 & 1 & 0 & 0 & 0 \\ 0 & 1 & 1 & 1 & 1 & 0 & 0 & 0 \\ 0 & 1 & 1 & 1 & 1 & 0 & 0 & 0 \\ 0 & 1 & 1 & 1 & 1 & 0 & 0 & 0 \\ 0 & 0 & 0 & 0 & 0 & 0 & 0 & 0 \\ 0 & 0 & 0 & 0 & 0 & 0 & 0 & 0 \end{bmatrix}$$

$$p = \begin{bmatrix} 0 & 0 & 0 & 0 & 0 & 1 & 1 & 0 \\ 0 & 0 & 0 & 0 & 0 & 1 & 1 & 0 \\ 0 & 0 & 0 & 0 & 0 & 0 & 0 & 0 \\ 0 & 0 & 1 & 0 & 0 & 0 & 0 & 0 \\ 0 & 0 & 1 & 1 & 0 & 0 & 0 & 0 \\ 0 & 0 & 1 & 1 & 0 & 0 & 0 & 0 \\ 0 & 0 & 0 & 0 & 0 & 0 & 1 & 1 \\ 0 & 0 & 0 & 0 & 0 & 0 & 1 & 0 \end{bmatrix}$$

$$\hat{p} = \begin{bmatrix} 0 & 0 & 0 & 0 & 0 & 0 & 0 & 0 \\ 0 & 0 & 0 & 0 & 0 & 0 & 0 & 0 \\ 0 & 0 & 0 & 0 & 0 & 0 & 0 & 0 \\ 0 & 0 & 1 & 0 & 0 & 0 & 0 & 0 \\ 0 & 0 & 1 & 1 & 0 & 0 & 0 & 0 \\ 0 & 0 & 1 & 1 & 0 & 0 & 0 & 0 \\ 0 & 0 & 0 & 0 & 0 & 0 & 1 & 1 \\ 0 & 0 & 0 & 0 & 0 & 0 & 1 & 0 \end{bmatrix}$$

Then $CA(g, p) = 16/25 = CA(g, \hat{p})$.

**Prop.6** Consider the following example.

$$g = \begin{bmatrix} 0 & 0 & 0 & 0 & 0 & 0 & 0 & 0 \\ 0 & 1 & 1 & 1 & 1 & 0 & 0 & 0 \\ 0 & 1 & 1 & 1 & 1 & 0 & 0 & 0 \\ 0 & 1 & 1 & 1 & 1 & 0 & 0 & 0 \\ 0 & 1 & 1 & 1 & 1 & 0 & 0 & 0 \\ 0 & 1 & 1 & 1 & 1 & 0 & 0 & 0 \\ 0 & 0 & 0 & 0 & 0 & 0 & 0 & 0 \\ 0 & 0 & 0 & 0 & 0 & 0 & 0 & 0 \end{bmatrix}$$

$$p = \begin{bmatrix} 0 & 0 & 0 & 0 & 0 & 1 & 1 & 0 \\ 0 & 0 & 0 & 0 & 0 & 1 & 1 & 0 \\ 0 & 0 & 0 & 0 & 0 & 0 & 0 & 0 \\ 0 & 0 & 0 & 0 & 0 & 0 & 0 & 0 \\ 0 & 0 & 1 & 1 & 1 & 0 & 0 & 0 \\ 0 & 0 & 1 & 1 & 1 & 0 & 0 & 0 \\ 0 & 0 & 1 & 1 & 0 & 0 & 0 & 0 \\ 0 & 0 & 0 & 0 & 0 & 0 & 0 & 0 \end{bmatrix}$$

$$\hat{p} = \begin{bmatrix} 0 & 0 & 0 & 0 & 0 & 1 & 1 & 0 \\ 0 & 0 & 0 & 0 & 0 & 1 & 1 & 0 \\ 0 & 0 & 0 & 0 & 0 & 0 & 0 & 0 \\ 0 & 0 & 0 & 0 & 0 & 0 & 0 & 0 \\ 0 & 0 & 1 & 1 & 1 & 0 & 0 & 0 \\ 0 & 0 & 1 & 1 & 1 & 0 & 0 & 0 \\ 0 & 0 & 0 & 0 & 0 & 0 & 0 & 0 \\ 0 & 0 & 1 & 1 & 0 & 0 & 0 & 0 \end{bmatrix}$$

Then $CA(g,p) = 67/100 = CA(g,\hat{p})$.

**Prop.7.** Let $a^* \in p^{-1}(1) \cap g^{-1}(1)$ and let $n^* \in p^{-1}(0) \cap g^{-1}(0)$. Define $q$ by the TP→FP shift

$$q^{-1}(1) = \left(p^{-1}(1) \setminus \{a^*\}\right) \cup \{n^*\}.$$

Then $\mathrm{CA}(g,p) > \mathrm{CA}(g,q)$.

The construction changes the prediction only at the two pixels $a^*$ and $n^*$.

1) Since $a^* \in p^{-1}(1) \cap g^{-1}(1)$, we have $p(a^*) = 1$ and $g(a^*) = 1$, so $p$ is correct at $a^*$. In $q$, the pixel $a^*$ is removed from the positive set, hence $q(a^*) = 0$, which is incorrect because $g(a^*) = 1$. Thus the number of correct pixels decreases by 1 due to $a^*$.

2) Since $n^* \in p^{-1}(0) \cap g^{-1}(0)$, we have $p(n^*) = 0$ and $g(n^*) = 0$, so $p$ is correct at $n^*$. In $q$, the pixel $n^*$ is added to the positive set, hence $q(n^*) = 1$, which is incorrect because $g(n^*) = 0$. Thus the number of correct pixels decreases by another 1 due to $n^*$.

All other pixels are unchanged, hence the total number of correct pixels decreases by 2:

$$|q^{-1}(1) \cap g^{-1}(1)| + |q^{-1}(0) \cap g^{-1}(0)| = \left(|p^{-1}(1) \cap g^{-1}(1)| + |p^{-1}(0) \cap g^{-1}(0)|\right) - 2.$$

Dividing by the constant $|D|$ yields $\mathrm{CA}(g,p) > \mathrm{CA}(g,q)$.

**Prop.8** Consider the following example:

$$g = \begin{bmatrix} 0 & 0 & 0 & 0 & 0 & 0 & 0 & 0 \\ 0 & 1 & 1 & 1 & 1 & 1 & 0 & 0 \\ 0 & 1 & 1 & 1 & 1 & 1 & 0 & 0 \\ 0 & 1 & 1 & 1 & 1 & 1 & 0 & 0 \\ 0 & 1 & 1 & 1 & 1 & 1 & 0 & 0 \\ 0 & 1 & 1 & 1 & 1 & 1 & 0 & 0 \\ 0 & 0 & 0 & 0 & 0 & 0 & 0 & 0 \\ 0 & 0 & 0 & 0 & 0 & 0 & 0 & 0 \end{bmatrix}$$

$$p = \begin{bmatrix} 0 & 0 & 0 & 0 & 0 & 0 & 0 & 0 \\ 0 & 0 & 0 & 0 & 0 & 0 & 0 & 0 \\ 0 & 0 & 0 & 0 & 0 & 0 & 0 & 0 \\ 0 & 0 & 1 & 1 & 1 & 0 & 0 & 0 \\ 0 & 0 & 1 & 1 & 0 & 0 & 0 & 0 \\ 0 & 0 & 0 & 0 & 0 & 0 & 0 & 0 \\ 0 & 0 & 0 & 0 & 0 & 0 & 0 & 0 \\ 0 & 0 & 0 & 0 & 0 & 0 & 0 & 0 \end{bmatrix}$$

$$\hat{p} = \begin{bmatrix} 0 & 0 & 0 & 0 & 0 & 0 & 0 & 0 \\ 0 & 1 & 1 & 0 & 0 & 0 & 0 & 0 \\ 0 & 1 & 0 & 0 & 0 & 0 & 0 & 0 \\ 0 & 1 & 0 & 0 & 0 & 0 & 0 & 0 \\ 0 & 1 & 0 & 0 & 0 & 0 & 0 & 0 \\ 0 & 0 & 0 & 0 & 0 & 0 & 0 & 0 \\ 0 & 0 & 0 & 0 & 0 & 0 & 0 & 0 \\ 0 & 0 & 0 & 0 & 0 & 0 & 0 & 0 \end{bmatrix}$$

Then $CA(g, p) = 3/5 = CA(g, \hat{p})$.

## B.6. Theorem 3.6 The Matthews Correlation Coefficient (MCC)

**Prop.1.** Let $A \in I_1(g)$ and let $p, q$ satisfy $p_{\mathrm{dom}(g) \setminus A} = q_{\mathrm{dom}(g) \setminus A}$ and $q_A = 0$. Assume $|p^{-1}(1) \cap A| > 0$ and MCC is defined for both $(g, p)$ and $(g, q)$. Then

$$\mathrm{MCC}(g, p) > \mathrm{MCC}(g, q).$$

Because $A \in I_1(g)$ is a ground-truth component, $A \subseteq g^{-1}(1)$. Since $q_A = 0$, $q$ predicts 0 on all pixels of $A$, so every pixel of $A$ predicted positive by $p$ becomes a false negative under $q$.

Let $\epsilon := |p^{-1}(1) \cap A| > 0$. Outside $A$, the predictions coincide, hence $FP$ and $TN$ are identical for $p$ and $q$, and only $TP$ and $FN$ differ:

$$TP(p) = TP(q) + \epsilon, \qquad FN(p) = FN(q) - \epsilon, \qquad FP(p) = FP(q), \qquad TN(p) = TN(q).$$

Write the counts of $q$ as $(TP_0, TN_0, FP_0, FN_0)$, so the counts of $p$ are $(TP_0 + \epsilon, TN_0, FP_0, FN_0 - \epsilon)$.

Consider the MCC numerator:

$$N(p) = (TP_0 + \epsilon)TN_0 - FP_0(FN_0 - \epsilon) = (TP_0 TN_0 - FP_0 FN_0) + \epsilon(TN_0 + FP_0).$$

Since $TN_0 + FP_0 = |g^{-1}(0)| > 0$ and $\epsilon > 0$, we have $N(p) > N(q)$.

All denominator factors remain strictly positive by the non-degeneracy assumption, so the denominator is positive for both predictions. Hence increasing the numerator strictly increases the MCC value:

$$\mathrm{MCC}(g, p) > \mathrm{MCC}(g, q).$$

**Prop.2** Consider the following example:

$$g = \begin{bmatrix} 0 & 0 & 0 & 0 & 0 & 0 & 0 & 0 \\ 0 & 1 & 1 & 1 & 1 & 1 & 0 & 0 \\ 0 & 1 & 1 & 1 & 1 & 1 & 0 & 0 \\ 0 & 1 & 1 & 1 & 1 & 1 & 0 & 0 \\ 0 & 1 & 1 & 1 & 1 & 1 & 0 & 0 \\ 0 & 1 & 1 & 1 & 1 & 1 & 0 & 0 \\ 0 & 0 & 0 & 0 & 0 & 0 & 0 & 0 \\ 0 & 0 & 0 & 0 & 0 & 0 & 0 & 0 \end{bmatrix}$$

$$p = \begin{bmatrix} 0 & 0 & 0 & 0 & 0 & 0 & 0 & 0 \\ 0 & 1 & 1 & 1 & 0 & 0 & 0 & 0 \\ 0 & 1 & 1 & 1 & 0 & 0 & 0 & 0 \\ 0 & 0 & 0 & 0 & 0 & 0 & 0 & 0 \\ 0 & 0 & 0 & 0 & 0 & 0 & 0 & 0 \\ 0 & 0 & 0 & 0 & 0 & 0 & 0 & 0 \\ 0 & 0 & 0 & 0 & 0 & 0 & 0 & 0 \\ 0 & 0 & 0 & 0 & 0 & 0 & 0 & 0 \end{bmatrix}$$

$$\hat{p} = \begin{bmatrix} 0 & 0 & 0 & 0 & 0 & 0 & 0 & 0 \\ 0 & 1 & 1 & 0 & 0 & 0 & 0 & 0 \\ 0 & 1 & 1 & 1 & 0 & 0 & 0 & 0 \\ 0 & 0 & 0 & 0 & 0 & 0 & 0 & 0 \\ 0 & 0 & 1 & 1 & 0 & 0 & 0 & 0 \\ 0 & 0 & 0 & 1 & 0 & 0 & 0 & 0 \\ 0 & 0 & 0 & 0 & 0 & 0 & 0 & 0 \\ 0 & 0 & 0 & 0 & 0 & 0 & 0 & 0 \end{bmatrix}$$

Then $MCC(g,p) = 2/5 < 1/2 = MCC(g,\hat{p})$.

**Prop.3 (MCC)** Let $n \in g^{-1}(0)$ and let $p, q$ satisfy $p_{\mathrm{dom}(g)\setminus\{n\}} = q_{\mathrm{dom}(g)\setminus\{n\}}$, $p(n) = 0$ and $q(n) = 1$. Assume MCC is defined for both $(g, p)$ and $(g, q)$. Then
$$\mathrm{MCC}(g, p) > \mathrm{MCC}(g, q).$$

Since $g(n) = 0$ and $p(n) = 0$, pixel $n$ contributes to $TN$ for $p$. After flipping to $q(n) = 1$, the same pixel contributes to $FP$ for $q$. All other pixels are unchanged, hence the confusion counts satisfy

$$TP(q) = TP(p), \quad FN(q) = FN(p), \quad TN(q) = TN(p) - 1, \quad FP(q) = FP(p) + 1.$$

Write $TP, TN, FP, FN$ for the counts of $p$, and define

$$N := TP \cdot TN - FP \cdot FN, \qquad D := \sqrt{(TP + FP)(TN + FN)(TP + FN)(TN + FP)}.$$

Then $\mathrm{MCC}(g, p) = N/D$.

For $q$ the numerator becomes

$$N' = TP(TN - 1) - (FP + 1)FN = (TP \cdot TN - FP \cdot FN) - (TP + FN) = N - (TP + FN),$$

so the numerator decreases by the strictly positive quantity $(TP + FN) = |g^{-1}(1)| > 0$.

For the denominator, note that $(TP + FN)$ and $(TN + FP)$ stay unchanged (ground-truth marginals), while the prediction marginals change as

$$TP + FP \mapsto TP + FP + 1, \qquad TN + FN \mapsto TN + FN - 1.$$

Hence $D' > 0$ and $\mathrm{MCC}(g, q) = N'/D'$ with $N' < N$. Since $D, D' > 0$, decreasing the numerator by a positive amount while keeping the denominator positive yields a strict decrease of the fraction, i.e. $\mathrm{MCC}(g, p) > \mathrm{MCC}(g, q)$.

**Prop.4** Consider the following example.

$$g = \begin{bmatrix} 0 & 0 & 0 & 0 & 0 & 0 & 0 & 0 \\ 0 & 1 & 1 & 1 & 1 & 0 & 0 & 0 \\ 0 & 1 & 1 & 1 & 1 & 0 & 0 & 0 \\ 0 & 1 & 1 & 1 & 1 & 0 & 0 & 0 \\ 0 & 1 & 1 & 1 & 1 & 0 & 0 & 0 \\ 0 & 1 & 1 & 1 & 1 & 0 & 0 & 0 \\ 0 & 0 & 0 & 0 & 0 & 0 & 0 & 0 \\ 0 & 0 & 0 & 0 & 0 & 0 & 0 & 0 \end{bmatrix}$$

$$p = \begin{bmatrix} 0 & 0 & 0 & 0 & 0 & 1 & 1 & 0 \\ 0 & 0 & 0 & 0 & 0 & 1 & 1 & 0 \\ 0 & 0 & 0 & 0 & 0 & 0 & 0 & 0 \\ 0 & 0 & 0 & 0 & 0 & 0 & 0 & 0 \\ 0 & 0 & 1 & 1 & 1 & 0 & 0 & 0 \\ 0 & 0 & 1 & 1 & 1 & 0 & 0 & 0 \\ 0 & 0 & 1 & 1 & 1 & 0 & 0 & 0 \\ 0 & 0 & 0 & 0 & 0 & 0 & 0 & 0 \end{bmatrix}$$

$$\hat{p} = \begin{bmatrix} 0 & 0 & 0 & 0 & 0 & 1 & 1 & 0 \\ 0 & 0 & 0 & 0 & 0 & 1 & 1 & 0 \\ 0 & 0 & 0 & 0 & 0 & 0 & 0 & 0 \\ 0 & 0 & 0 & 0 & 0 & 0 & 0 & 0 \\ 0 & 0 & 1 & 1 & 1 & 0 & 0 & 0 \\ 0 & 0 & 1 & 1 & 1 & 0 & 0 & 0 \\ 0 & 0 & 0 & 0 & 0 & 1 & 0 & 0 \\ 0 & 0 & 1 & 1 & 0 & 0 & 0 & 0 \end{bmatrix}$$

Then $MCC(g, p) = 23/100 = MCC(g, \hat{p})$.

**Prop.5** Consider the following example:

$$g = \begin{bmatrix} 0 & 0 & 0 & 0 & 0 & 0 & 0 & 0 \\ 0 & 1 & 1 & 1 & 1 & 0 & 0 & 0 \\ 0 & 1 & 1 & 1 & 1 & 0 & 0 & 0 \\ 0 & 1 & 1 & 1 & 1 & 0 & 0 & 0 \\ 0 & 1 & 1 & 1 & 1 & 0 & 0 & 0 \\ 0 & 1 & 1 & 1 & 1 & 0 & 0 & 0 \\ 0 & 0 & 0 & 0 & 0 & 0 & 0 & 0 \\ 0 & 0 & 0 & 0 & 0 & 0 & 0 & 0 \end{bmatrix}$$

$$p = \begin{bmatrix} 0 & 0 & 0 & 0 & 0 & 1 & 1 & 0 \\ 0 & 0 & 0 & 0 & 0 & 1 & 1 & 0 \\ 0 & 0 & 0 & 0 & 0 & 0 & 0 & 0 \\ 0 & 0 & 1 & 0 & 0 & 0 & 0 & 0 \\ 0 & 0 & 1 & 1 & 0 & 0 & 0 & 0 \\ 0 & 0 & 1 & 1 & 0 & 0 & 0 & 0 \\ 0 & 0 & 0 & 0 & 0 & 0 & 1 & 1 \\ 0 & 0 & 0 & 0 & 0 & 0 & 1 & 0 \end{bmatrix}$$

$$\hat{p} = \begin{bmatrix} 0 & 0 & 0 & 0 & 0 & 0 & 0 & 0 \\ 0 & 0 & 0 & 0 & 0 & 0 & 0 & 0 \\ 0 & 0 & 0 & 0 & 0 & 0 & 0 & 0 \\ 0 & 0 & 1 & 0 & 0 & 0 & 0 & 0 \\ 0 & 0 & 1 & 1 & 0 & 0 & 0 & 0 \\ 0 & 0 & 1 & 1 & 0 & 0 & 0 & 0 \\ 0 & 0 & 0 & 0 & 0 & 0 & 1 & 1 \\ 0 & 0 & 0 & 0 & 0 & 0 & 1 & 0 \end{bmatrix}$$

Then $MCC(g,p) = 9/50 = MCC(g,\hat{p})$.

**Prop.6** Consider the following example:

$$g = \begin{bmatrix} 0 & 0 & 0 & 0 & 0 & 0 & 0 & 0 \\ 0 & 1 & 1 & 1 & 1 & 0 & 0 & 0 \\ 0 & 1 & 1 & 1 & 1 & 0 & 0 & 0 \\ 0 & 1 & 1 & 1 & 1 & 0 & 0 & 0 \\ 0 & 1 & 1 & 1 & 1 & 0 & 0 & 0 \\ 0 & 1 & 1 & 1 & 1 & 0 & 0 & 0 \\ 0 & 0 & 0 & 0 & 0 & 0 & 0 & 0 \\ 0 & 0 & 0 & 0 & 0 & 0 & 0 & 0 \end{bmatrix}$$

$$p = \begin{bmatrix} 0 & 0 & 0 & 0 & 0 & 1 & 1 & 0 \\ 0 & 0 & 0 & 0 & 0 & 1 & 1 & 0 \\ 0 & 0 & 0 & 0 & 0 & 0 & 0 & 0 \\ 0 & 0 & 0 & 0 & 0 & 0 & 0 & 0 \\ 0 & 0 & 1 & 1 & 1 & 0 & 0 & 0 \\ 0 & 0 & 1 & 1 & 1 & 0 & 0 & 0 \\ 0 & 0 & 1 & 1 & 0 & 0 & 0 & 0 \\ 0 & 0 & 0 & 0 & 0 & 0 & 0 & 0 \end{bmatrix}$$

$$\hat{p} = \begin{bmatrix} 0 & 0 & 0 & 0 & 0 & 1 & 1 & 0 \\ 0 & 0 & 0 & 0 & 0 & 1 & 1 & 0 \\ 0 & 0 & 0 & 0 & 0 & 0 & 0 & 0 \\ 0 & 0 & 0 & 0 & 0 & 0 & 0 & 0 \\ 0 & 0 & 1 & 1 & 1 & 0 & 0 & 0 \\ 0 & 0 & 1 & 1 & 1 & 0 & 0 & 0 \\ 0 & 0 & 0 & 0 & 0 & 0 & 0 & 0 \\ 0 & 0 & 1 & 1 & 0 & 0 & 0 & 0 \end{bmatrix}$$

Then $MCC(g,p) = 7/50 = MCC(g,\hat{p})$.

**Prop.7.** Let $a^* \in p^{-1}(1) \cap g^{-1}(1)$ and $n^* \in p^{-1}(0) \cap g^{-1}(0)$. Define $q$ by the TP→FP shift: $q(a^*) = 0$, $q(n^*) = 1$, and $q = p$ elsewhere. Assume MCC is defined for both $(g,p)$ and $(g,q)$. Then

$$\mathrm{MCC}(g,p) > \mathrm{MCC}(g,q).$$

At $a^*$ we change a correct positive into an incorrect negative, so $TP$ decreases by $1$ and $FN$ increases by $1$. At $n^*$ we change a correct negative into an incorrect positive, so $TN$ decreases by $1$ and $FP$ increases by $1$. All other pixels are unchanged, hence

$$TP(q) = TP - 1, \quad FN(q) = FN + 1, \quad TN(q) = TN - 1, \quad FP(q) = FP + 1.$$

Let $N := TP \cdot TN - FP \cdot FN$ be the numerator for $p$. For $q$ the numerator equals

$$
\begin{aligned}
N' &= (TP - 1)(TN - 1) - (FP + 1)(FN + 1) \\
&= TP \cdot TN - TP - TN + 1 - (FP \cdot FN + FP + FN + 1) \\
&= (TP \cdot TN - FP \cdot FN) - (TP + TN + FP + FN) \\
&= N - |\operatorname{dom}(g)|.
\end{aligned}
$$

Thus the numerator decreases by $|\operatorname{dom}(g)| > 0$.

For the denominator, all four factors remain positive under the non-degeneracy assumption, so the denominator stays strictly positive. Therefore $\mathrm{MCC}(g, q) = N'/D'$ with $N' < N$ and $D' > 0$, which implies $\mathrm{MCC}(g, p) > \mathrm{MCC}(g, q)$.

**Prop.8** Consider the following example:

$$
g = \begin{bmatrix}
0 & 0 & 0 & 0 & 0 & 0 & 0 & 0 \\
0 & 1 & 1 & 1 & 1 & 1 & 0 & 0 \\
0 & 1 & 1 & 1 & 1 & 1 & 0 & 0 \\
0 & 1 & 1 & 1 & 1 & 1 & 0 & 0 \\
0 & 1 & 1 & 1 & 1 & 1 & 0 & 0 \\
0 & 1 & 1 & 1 & 1 & 1 & 0 & 0 \\
0 & 0 & 0 & 0 & 0 & 0 & 0 & 0 \\
0 & 0 & 0 & 0 & 0 & 0 & 0 & 0
\end{bmatrix}
$$

$$
p = \begin{bmatrix}
0 & 0 & 0 & 0 & 0 & 0 & 0 & 0 \\
0 & 0 & 0 & 0 & 0 & 0 & 0 & 0 \\
0 & 0 & 0 & 0 & 0 & 0 & 0 & 0 \\
0 & 0 & 1 & 1 & 1 & 0 & 0 & 0 \\
0 & 0 & 1 & 1 & 1 & 0 & 0 & 0 \\
0 & 0 & 0 & 1 & 1 & 0 & 0 & 0 \\
0 & 0 & 0 & 0 & 0 & 0 & 0 & 0 \\
0 & 0 & 0 & 0 & 0 & 0 & 0 & 0
\end{bmatrix}
$$

$$
\hat{p} = \begin{bmatrix}
0 & 0 & 0 & 0 & 0 & 0 & 0 & 0 \\
0 & 1 & 1 & 1 & 1 & 0 & 0 & 0 \\
0 & 1 & 0 & 0 & 0 & 0 & 0 & 0 \\
0 & 1 & 0 & 0 & 0 & 0 & 0 & 0 \\
0 & 1 & 0 & 0 & 0 & 0 & 0 & 0 \\
0 & 0 & 0 & 0 & 0 & 0 & 0 & 0 \\
0 & 0 & 0 & 0 & 0 & 0 & 0 & 0 \\
0 & 0 & 0 & 0 & 0 & 0 & 0 & 0
\end{bmatrix}
$$

Then $MCC(g, p) = 2/5 = MCC(g, \hat{p})$.

## C. Proofs of the Properties of SAAM-ALARM

Recall that

$$
m(g, p) = \frac{1}{2} \left[ \frac{1}{N_G} \sum_{i=1}^{N_G} \frac{\alpha_i(p) + 1}{b^{\max(0, n_i(p) - 1)}} - \sum_{P_j \in \mathcal{F}(p)} \omega(P_j)\big(\lambda + \beta(P_j)\big) \right],
$$

where $b > 1$, $\lambda > 0$, $\eta > 0$, and $k > 0$.

**Proposition C.1** (Property 1). *SAAM-ALARM satisfies Property 1.*

Let $A = A_i \in I_1(g)$ and let $p, q$ satisfy $p_{\operatorname{dom}(g) \setminus A} = q_{\operatorname{dom}(g) \setminus A}$ and $q_A \equiv 0$. Since $A \subset g^{-1}(1)$, we have

$$
\alpha_i(q) = 0.
$$

Because $A \in TD_1(p,g)$, there exists $P \in I_1(p)$ such that $A \cap P \neq \emptyset$, hence $\alpha_i(p) > 0$. For all $\ell \neq i$,

$$\alpha_\ell(p) = \alpha_\ell(q), \qquad n_\ell(p) = n_\ell(q),$$

and $\mathcal{F}(p) = \mathcal{F}(q)$. Therefore

$$m(g,p) - m(g,q) = \frac{1}{2N_G}\left(\frac{\alpha_i(p)+1}{b^{\max(0,n_i(p)-1)}} - 1\right) > 0.$$

**Proposition C.2** (Property 2). *SAAM-ALARM satisfies Property 2.*

Let $A_i \in TD_1(p,g)$ and let $q$ be constructed as in Property 2. Then $\alpha_i(q) \leq \alpha_i(p)$ and

$$n_i(q) = n_i(p) + 1.$$

Hence

$$\frac{\alpha_i(q)+1}{b^{\max(0,n_i(q)-1)}} \leq \frac{\alpha_i(p)+1}{b^{\max(0,n_i(p))}} < \frac{\alpha_i(p)+1}{b^{\max(0,n_i(p)-1)}}.$$

All other reward terms and all penalty terms are unchanged. Thus $m(g,p) > m(g,q)$.

**Proposition C.3** (Property 3). *SAAM-ALARM satisfies Property 3.*

Let $n \in g^{-1}(0)$ and let $q$ be defined by $q(n) = 1$ and $q = p$ on $\mathrm{dom}(g) \setminus \{n\}$. Then $\alpha_i(q) = \alpha_i(p)$ for all $i$. Either a new false-positive component $P$ is created or an existing false-positive component grows. In both cases,

$$\sum_{P_j \in \mathcal{F}(q)} \omega(P_j)\big(\lambda + \beta(P_j)\big) > \sum_{P_j \in \mathcal{F}(p)} \omega(P_j)\big(\lambda + \beta(P_j)\big),$$

since $\lambda > 0$ and $\beta$ is strictly increasing in $|P_j|$. Therefore $m(g,p) > m(g,q)$.

**Proposition C.4** (Property 4). *SAAM-ALARM satisfies Property 4.*

Let $p, q$ satisfy $p_{g=1} = q_{g=1}$ and $I_1(q_{g=0}) = I_1(p_{g=0}) \sqcup A$. Then $\alpha_i(p) = \alpha_i(q)$ and $n_i(p) = n_i(q)$ for all $i$. Moreover, $\mathcal{F}(q) = \mathcal{F}(p) \sqcup \{A\}$. Thus

$$m(g,p) - m(g,q) = \frac{1}{2}\omega(A)\big(\lambda + \beta(A)\big) > 0.$$

**Proposition C.5** (Property 5). *SAAM-ALARM satisfies Property 5.*

Let $A_p \in I_1(p)$ and $A_q \in I_1(q)$ be false-positive components with $|A_p| = |A_q|$ and $d(A_p, \mathcal{A}) < d(A_q, \mathcal{A})$. Then $\beta(A_p) = \beta(A_q)$ and

$$\omega(A_p) < \omega(A_q).$$

All reward terms and all other penalty terms coincide. Hence

$$m(g,p) - m(g,q) = \frac{1}{2}\big(\omega(A_q) - \omega(A_p)\big)\big(\lambda + \beta(A_p)\big) > 0.$$

**Proposition C.6** (Property 6). *SAAM-ALARM satisfies Property 6 provided that $\omega_{\min}\lambda > \frac{1}{N_G}$.*

Let $q$ be constructed from $p$ as in Property 6. Then for the affected ground-truth component $A_i$,

$$\alpha_i(q) \geq \alpha_i(p),$$

and the maximal increase of the reward term is bounded by

$$\frac{1}{2N_G}.$$

Under $q$, a new false-positive component $Q$ is created, yielding an additional penalty of at least

$$\frac{1}{2}\omega_{\min}\lambda.$$

Thus

$$m(g,p) - m(g,q) \geq \frac{1}{2}\omega_{\min}\lambda - \frac{1}{2N_G} > 0.$$

**Proposition C.7** (Property 7). *SAAM-ALARM satisfies Property 7.*

Let $a^* \in p^{-1}(1) \cap g^{-1}(1)$ and $n^* \in p^{-1}(0) \cap g^{-1}(0)$, and let $q$ be obtained by the TP→FP shift. Then for the affected ground-truth component $A_i$,

$$\sum_{u \in A_i \cap P_i^*(q)} W_i(u) < \sum_{u \in A_i \cap P_i^*(p)} W_i(u),$$

and $\mathrm{IoU}(A_i, P_i^*(q)) \leq \mathrm{IoU}(A_i, P_i^*(p))$. Hence $\alpha_i(q) < \alpha_i(p)$ and

$$\frac{\alpha_i(q) + 1}{b^{\max(0, n_i(q)-1)}} < \frac{\alpha_i(p) + 1}{b^{\max(0, n_i(p)-1)}}.$$

All other terms do not increase, therefore $m(g, p) > m(g, q)$.

**Proposition C.8** (Property 8). *SAAM-ALARM satisfies Property 8.*

Let $P, Q \subset A_i$ with $|P| = |Q|$ and

$$\sum_{a \in P} \delta_{A_i}(a) > \sum_{a \in Q} \delta_{A_i}(a).$$

Since $W_i$ is strictly increasing in centrality,

$$\sum_{a \in P} W_i(a) > \sum_{a \in Q} W_i(a).$$

Moreover, $\mathrm{IoU}(A_i, P) = \mathrm{IoU}(A_i, Q)$. Thus $\alpha_i(P) > \alpha_i(Q)$ and all other terms coincide, implying $m(g, p) > m(g, q)$.

**Theorem C.9.** *Under the stated parameter assumptions, SAAM-ALARM satisfies Properties P1–P8.*

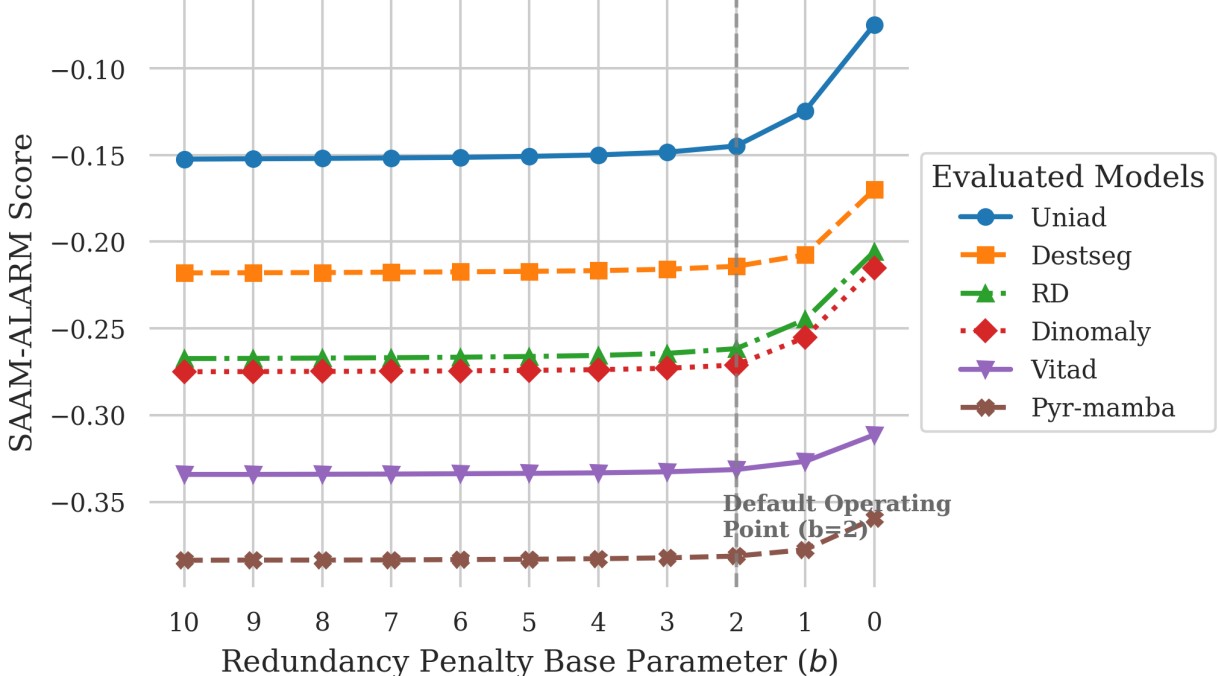

*Figure 6.* Sensitivity analysis of the redundancy penalty base $b$ alongside standard pixel-level metrics across six distinct anomaly detection models. While absolute SAAM-ALARM scores exhibit graceful saturation as $b$ increases, the relative performance ranking of the models remains completely invariant, demonstrating robust metric stability.

## D. Hyperparameter Sensitivity Analysis

**Empirical Analysis of Parameter** $b$    As illustrated in Figure 6, we evaluate the behavior of SAAM-ALARM across an extensive parameter sweep $b \in [0, 10]$ across six foundational models.

Two critical phenomena emerge from this analysis: First, the relative ranking of the models is completely invariant to changes in $b$. Whether the redundancy penalty is mild ($b = 1$) or exceptionally aggressive ($b = 10$), the ordering remains rigidly preserved (UniAD $\succ$ Destseg $\succ$ RD $\succ$ Dinomaly $\succ \dots$). This provides definitive empirical verification that the metric evaluates systemic architectural traits rather than artifacts of hyperparameter configurations.

Second, the data highlights that the most significant optimization and dynamic variation happen when $b \leq 2$. Beyond $b = 3$, the penalty function hits a stable saturation plateau where marginal shifts in absolute scores become negligible. This perfectly justifies our selection of $b = 2$ as a well-calibrated baseline operating point that effectively punishes redundant bounding boxes without inducing premature signal degradation.

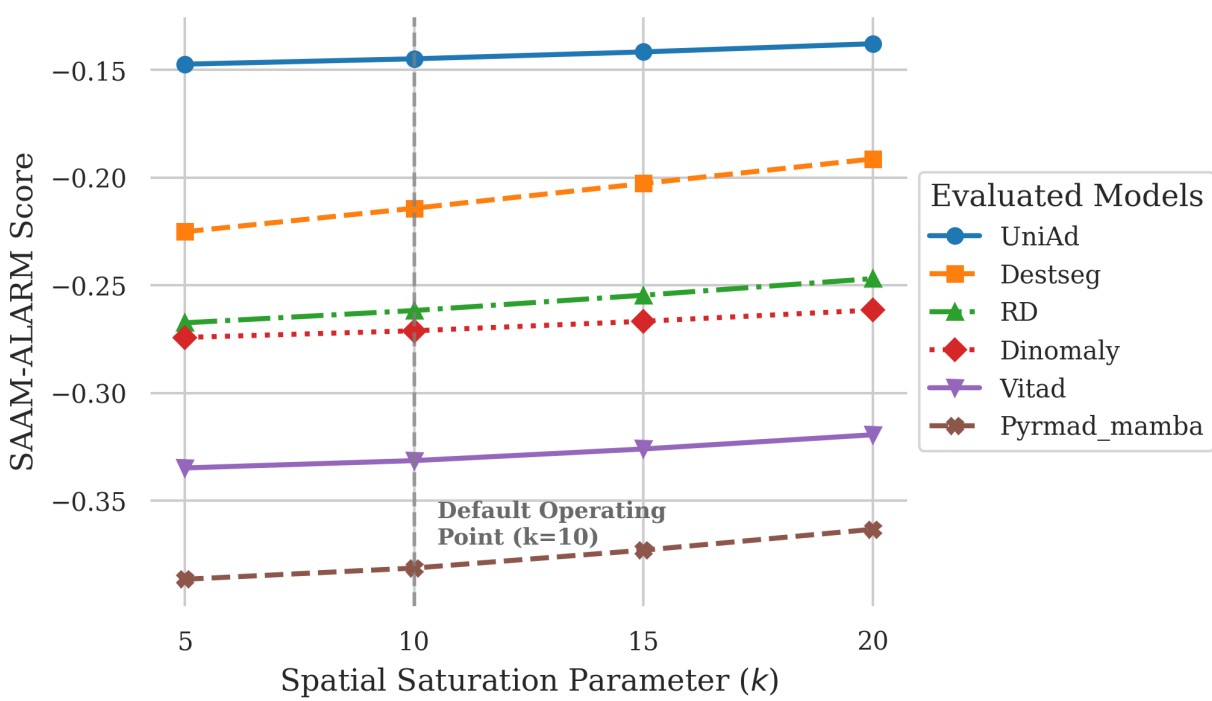

**Hyperparameter Sensitivity Analysis ($k$ Summary Sweep)**

*Figure 7.* Sensitivity analysis of the spatial saturation hyperparameter $k \in \{5, 10, 15, 20\}$ across six unsupervised AD models. As $k$ increases, absolute SAAM-ALARM scores exhibit a uniform, gradual upward shift because a wider spatial penalty distribution naturally reduces strictness toward distant false positives.

**Empirical Analysis of Parameter** $k$    The proximity-weighted false positive saturation parameter $k$ is evaluated across a discrete sweep $k \in \{5, 10, 15, 20\}$, as detailed in the appendix visualizations. As theoretically expected, increasing the value of $k$ shifts the absolute SAAM-ALARM scores slightly upward across all six benchmarks. This uniform behavior occurs because a larger $k$ broadens the spatial penalty distribution, effectively softening the penalty assigned to distant false positives.

Figure 7 demonstrates that despite the absolute score variations, the model performance hierarchy remains entirely invariant across the entire evaluated range: UniAd $\succ$ Destseg $\succ$ RD $\succ$ Dinomaly $\succ$ Vitad $\succ$ Pyrmad_mamba. This global monotonicity proves that the metric provides a structurally robust assessment that does not artificially manipulate relative model benchmarking outcomes. Our selection of $k = 10$ serves as a balanced, interpretable middle ground providing adequate spatial discrimination without oversaturating localization errors.

# E. Generalization and Evaluation on the VisA Dataset

*Table 2.* Model ranking variance on the VisA dataset across evaluation metrics. Traditional metrics yield inconsistent model orderings due to insufficient penalization of redundant predictions, whereas SAAM produces a more consistent ranking by considering spatial overlap and redundancy.

| Metric | UniAD | P-Mamba | Vitad | Resulting Ranking |
|---|---|---|---|---|
| Precision | 0.1781 | 0.2123 | 0.1949 | P-Mamba > Vitad > UniAD |
| Recall | 0.2208 | 0.5068 | 0.3974 | P-Mamba > Vitad > UniAD |
| F1 | 0.1369 | 0.2281 | 0.2024 | P-Mamba > Vitad > UniAD |
| IOU | 0.0926 | 0.1525 | 0.1385 | P-Mamba > Vitad > UniAD |
| Accuracy | 0.9884 | 0.9835 | 0.9873 | UniAD > P-mamba > Vitad |
| Mcc | 0.1562 | 0.2706 | 0.2337 | P-Mamba > Vitad > UniAD |
| SAAM-ALARM | -0.0146 | -0.0261 | -0.0065 | Vitad > Uniad > P-mamba |

To demonstrate that the limitations of traditional metrics and the advantages of SAAM-ALARM are not specific to the MVTec-AD dataset, we provide an extensive evaluation on the full VisA dataset (Zou et al., 2022).

## E.1. Dataset Characteristics and Contamination Factor

The VisA dataset presents a highly challenging evaluation scenario consisting of 10,821 images across 12 distinct object classes. Crucially, we evaluate on the full dataset without any artificial class rebalancing or subsampling to preserve realistic testing conditions.

Among the test set, 1,200 images contain anomalies, yielding a class-level contamination factor of approximately 11.09%. At the pixel level, anomalous pixels constitute only 0.9987% of the entire dataset. This extreme sparsity reflects real-world deployment challenges where anomalies are localized and rare, making evaluation metrics highly sensitive to false positives, spatial localization deviations, and redundant regional predictions.

## E.2. Anomaly Detectors and Behavioral Profiles

We benchmark three state-of-the-art unsupervised anomaly detection models spanning different architectural paradigms: **UniAD**, **P-Mamba**, and **ViTAD**. All models are trained strictly on anomaly-free samples. As summarized in Table 2, these models exhibit highly distinct behavioral profiles on VisA:

- **Redundancy:** P-Mamba exhibits the highest redundancy rate (0.17), followed by UniAD (0.11) and ViTAD (0.08).

- **Spatial Accuracy:** The average Euclidean distance between the centroids of predicted anomalies and their corresponding ground truth targets is 6.0 pixels for ViTAD, 6.6 pixels for P-Mamba, and 7.8 pixels for UniAD.

- **False Positives:** P-Mamba generates the highest volume of false positives, while UniAD produces the lowest.

## E.3. Ranking Inconsistencies of Standard Metrics

Standard evaluation protocols relying on pixel-level overlap (Precision, Recall, Intersection-over-Union, and Accuracy) fail to comprehensively weigh these behavioral nuances. For instance, as shown in Table 2, standard Recall favors P-Mamba because its dense, redundant pixel predictions happen to cover ground-truth regions, despite its severe spatial inaccuracy and high false positive rate. Conversely, accuracy favors UniAD primarily due to its low false-positive footprint, failing to penalize its larger average spatial distance (7.8 pixels).

By explicitly accounting for the trade-offs between localization distance, spatial redundancy, and false positive penalties, SAAM-ALARM provides a stable, unified ranking of ViTAD $\succ$ UniAD $\succ$ P-Mamba. This aligns with an intuitive qualitative assessment of model reliability.

## E.4. Thresholding Strategy and Stability

All evaluations utilize pixel-level anomaly maps converted to binary masks via an adaptive thresholding strategy. The thresholds are computed per asset class based strictly on the score distributions of normal validation samples as $threshold =$

$\mu_{normal} + k * \sigma_{nomral}$. We use $k = 3$ for MVTec-AD and $k = 7$ for VisA, selected empirically to control false positives based on the score distributions of each dataset.

We employ an empirical threshold selection strategy designed to tightly control baseline false positives while adapting to cross-dataset variations. The baseline statistics of normal anomaly scores are highly consistent across both benchmarks (e.g., matching distributions observed between VisA and MVTec-AD). This confirms that our thresholding mechanism primarily calibrates detection strictness uniformly. Importantly, this thresholding protocol is held completely invariant across all evaluated metrics and detectors to guarantee a fair comparison. Furthermore, we empirically verified that the model rankings assigned by SAAM-ALARM remain perfectly robust across a wide perturbation range of these threshold values.

