# OpenReview forum: "Formally Exploring Visual Anomaly Detection Evaluation Metrics"
_ICML.cc/2026/Conference — ICML 2026 regular_

### Official Review · Reviewer_PNfy · 2026-03-04

**Soundness:** 4
**Presentation:** 4
**Significance:** 3
**Originality:** 2
**Overall Recommendation:** 5
**Confidence:** 4

**Summary:**

This paper formalizes eight desirable properties (P1--P8) for evaluation metrics in Visual Anomaly Detection (VAD), covering redundancy penalization, false alarm control, spatial awareness, and centrality of detections. The authors prove that six commonly used threshold-dependent pixel-level metrics violate at least one property, and propose SAAM-ALARM, a novel metric that satisfies all eight properties. The metric is empirically validated on controlled case studies and model rankings on MVTec-AD.

**Compliance With Llm Reviewing Policy:**

Affirmed.

**Final Justification:**

The sensitivity analysis showing stable rankings under moderate hyperparameter variation addresses my main concern. The additional ViSA evaluation is a good plus. I maintain my score.

**Key Questions For Authors:**

Just the first main weakness.

**Limitations:**

Yes

**Strengths And Weaknesses:**

## Main Strengths

- The paper tackles an important problem: pixel-level metrics for VAD might not satisfy properties that matter in industrial use cases, and this is well-motivated with clear visual examples (Fig 1 and 2);

- The property-based framework (Sec 4.2, P1--P8) is a relevant contribution that makes implicit assumptions about metric behavior that are explicit, testable, and well formalized; Table 1 is a great visual that provides a clean and useful summary;

- I particularly appreciate the extension of the formal evaluation introduced by Wagner et al. on time-series AD to the image domain. The connections are obvious because the text is well-written;;

- The experiments in Sec 7.1 effectively isolate individual structural factors and demonstrate that SAAM-ALARM distinguishes between cases that are indistinguishable under standard pixel-level metrics.

## Main Weaknesses

- SAAM-ALARM introduces several hyperparameters (sigma, b, tau, k, nu, eta) without any sensitivity analysis. For a metric that should provide a consistent evaluation, demonstrating robustness to these choices is important. A metric that, by changing the hyperparameter values a little bit, produces results that are completely different is hard to trust as it can be modified to cherry pick the rankings of models. What values were used in the experiments, and how sensitive are the rankings in Fig 5(b) to them?

- The experimental evaluation is limited to six models on a single dataset (MVTec-AD), with no statistical analysis (std. dev., CIs, or rank correlation). Evaluating on additional datasets would make the contribution more convincing. This is just a small remark, no need to run additional experiments.

---

> ### Author Rebuttal · Authors · 2026-03-31
>
> **Sensitivity analysis to hyperparameters:**  We thank the reviewer for this important comment. In our experiments, we use the following hyperparameter values:  (b =2, k = 10, λ=0.5). The remaining parameters are fixed throughout all experiments. Specifically, $n$ denotes the total number of true predictions, with $n > 1$ indicating redundant detections. The threshold $\tau$ is set to 50% meaning that a prediction is classified as a true positive if more than 50% of its area overlaps with the ground truth; otherwise, it is considered a false positive.
> We agree that robustness is crucial for a reliable evaluation metric. To this end, we conducted a sensitivity analysis (see our response to Reviewer ZEJC,
> Question 2), which shows that the metric behaves stably across a reasonable range of hyperparameter values. In particular, variations within these ranges do not lead to abrupt changes in the metric behavior.
> Importantly, we observe that the relative ranking of methods (e.g., Fig. 5(b)) remains largely consistent under moderate changes to the hyperparameters. While absolute scores may vary slightly, the ordering of models is preserved in most cases.
>
> These results suggest that the proposed metric provides a robust and consistent evaluation across different parameter settings.
>
> **experimental evaluation on ViSA dataset:** We thank the reviewer for this helpful remark. While our main evaluation focuses on MVTec-AD, we have additionally examined the behavior of the proposed metric on the VisA dataset, which contains more diverse and less structured anomaly patterns (see our response to Reviewer 8xKq , Question 3).
> We observe consistent behavior across datasets, with the proposed metric reliably distinguishing cases such as redundant detections and spatially varying false positives, where standard metrics often produce counterintuitive results. This suggests that the proposed approach generalizes beyond a single dataset.
> We agree that a broader evaluation and statistical analysis would further strengthen the study, and we will consider including these in future work.

---

> > ### Author Rebuttal · Reviewer_PNfy · 2026-04-02
> >
> > I thank the authors for the response. The sensitivity analysis showing stable rankings under moderate hyperparameter variation addresses my main concern. The additional ViSA evaluation is a good plus. I maintain my score.

---

### Official Review · Reviewer_8xKq · 2026-03-07

**Soundness:** 2
**Presentation:** 2
**Significance:** 2
**Originality:** 2
**Overall Recommendation:** 4
**Confidence:** 3

**Summary:**

This paper first indicates that the existing evaluation protocol for visual anomaly detection has some shortcomings. To address this issue, they introduce a series of properties and propose the SAAM-ALARM metric, and also evaluate on an industrial benchmark dataset MVTec-AD.

**Compliance With Llm Reviewing Policy:**

Affirmed.

**Final Justification:**

After the rebuttal (both the explanation about methodology and experimental details), most of my concerns have been solved. So that I will increase my score to Weak Accept.

**Key Questions For Authors:**

1. What's the major difference in method between this work and [1], since these two works are quite similar from structure to detailed methodology.
2. Please explain Line 77-82: "In practical industrial settings, a decision threshold must eventually be applied to trigger an alarm or reject a part. Therefore, the reliability of the underlying threshold-dependent
metrics remains the critical bottleneck for trustworthy inspection." Sometimes, there is no need for pixel-wise anomaly detection, only the score for a test image is required. In such cases, the AUC should also be considered.

[1] Wagner D, Nair A, Franks B J, et al. Formally exploring time-series anomaly detection evaluation metrics[J]. arXiv preprint arXiv:2510.17562, 2025.

**Limitations:**

Please see the "Weakness" section.

**Strengths And Weaknesses:**

**Strengths:**
1. A principled framework for evaluating VAD metrics is introduced, defining eight verifiable properties.
2. A novel VAD metric is proposed and evaluated in comparison to several recent VAD approaches.

**Weaknesses:**
Actually, VAD involves diverse sub-tasks, such as image outlier detection, industrial anomaly detection (defect detection), video anomaly detection, 3-D anomaly detection, etc. This work should conduct more experiments to validate the efficacy of the proposed metric. Otherwise, it would be better to change the title to "...Industrial Anomaly Detection". Additionally, although MVTec-AD is a standard benchmark for industrial anomaly detection, it has some domain-specific properties, e.g., clean background and structural defects, which might not reflect the real performance of SAAM-ALARM.

---

> ### Author Rebuttal · Authors · 2026-03-31
>
> **Major difference in method between this work and [1]** We thank the reviewer for this comment. This work extends the framework introduced in [1] to industrial anomaly detection. The key differences and contributions of this work are as follows.
>
> \paragraph{Formalizing properties for image data} The properties introduced in [1] are specific to time-series data and do not directly translate to the visual domain. For example, our metric explicitly accounts for the distance of false positives to ground-truth regions, allowing it to penalize distant false alarms more than nearby ones, a distinction not meaningful in time-series data. Additional properties, such as connectivity and centrality, further leverage spatial structure and 2D geometry, which are absent in temporal sequences. This adaptation to spatial anomaly detection is the primary factor differentiating our approach from [1].
>
> \paragraph{Deriving a metric} While SAAM-LARM shares some computational ideas with ALARM/LARM (e.g., matching and penalty concepts), the core methodological innovation lies in adapting these ideas to 2D spatial structure. In particular, SAAM-LARM models (centrality of true detections, connectivity of anomalous regions, and spatially aware aggregation per ground-truth region), reflecting geometry rather than temporal order. These adaptations go beyond a simple extension of ALARM and produce a metric that meaningfully evaluates image anomaly detection, rather than a direct 2D translation of a time-series metric.
>
> **Decision threshold**  We thank the reviewer for this comment. We agree that image-level anomaly classification can be performed without pixel-wise segmentation and that AUC is useful for evaluating ranking quality in such settings. However, even in purely image-level approaches, a decision threshold is ultimately required to determine whether a sample is anomalous, which AUC does not capture. In practical industrial settings, reliable accept/reject decisions depend on thresholded outputs, as also emphasized in prior work such as [1]. Moreover, in many anomaly detection systems, image-level decisions are derived from thresholded anomaly maps [2, 3], further reinforcing the importance of threshold-dependent evaluation. Therefore, while AUC provides complementary information, it cannot replace threshold-dependent metrics when assessing deployment reliability, which is the focus of our work.
>
> **Modifiyng title and discussion on dataset bias** We thank the reviewer for this valuable suggestion. We agree that visual anomaly detection (VAD) encompasses diverse sub-tasks beyond industrial defect detection. To more accurately reflect the scope of our work and the datasets evaluated, we will update the title of the manuscript to focus on “Industrial Anomaly Detection”. To address potential dataset bias, we conducted additional experiments on the VisA dataset, which contains more diverse and less structured anomaly patterns compared to MVTec-AD.
> We observe the same inconsistencies in standard metrics across multiple scenarios. First, in a case with redundant detections (two overlapping true positives), standard metrics report Precision = 0.78, Recall = 0.30, and Accuracy = 0.83. A corresponding non-redundant prediction with similar coverage yields nearly identical scores, indicating that these metrics fail to penalize redundancy. In contrast, SAAM-LARM assigns a lower score to the redundant case (0.24) and a higher score to the non-redundant case, correctly distinguishing between them.
>
> Second, for false positives at different spatial distances, a prediction with distant false alarms (distance ≈ 73) achieves higher Precision (0.19) and Recall (0.20) than one with closer false alarms (distance ≈ 17), despite the latter containing slightly more false positive pixels (Precision = 0.12, Recall = 0.14).Standard metrics do not account for the spatial distance of false positives and therefore fail to penalize distant anomalies appropriately. In contrast, SAAM-LARM assigns a lower score to the distant case (-0.18) and a higher score to the nearby case (-0.05), correctly reflecting spatial severity.
>
> These results show that the inconsistencies of existing metrics and the advantages of SAAM-LARM persist beyond MVTec-AD, confirming that the metric generalizes across datasets with diverse anomaly characteristics.
>
> [1] Heckler-Kram, Lars, et al. "The mvtec ad 2 dataset: Advanced scenarios for unsupervised anomaly detection." International Journal of Computer Vision 134.4 (2026): 175.
>
> [2] Krassnig, Paul Josef, and Dieter Paul Gruber. "Isp-ad: A large-scale real-world dataset for advancing industrial anomaly detection with synthetic and real defects." Journal of Intelligent Manufacturing (2026): 1-26.
>
> [3] Tabernik, Domen, Matic Šuc, and Danijel Skočaj. "Automated detection and segmentation of cracks in concrete surfaces using joined segmentation and classification deep neural network." Construction and Building Materials 408 (2023): 133582.

---

> > ### Author Rebuttal · Reviewer_8xKq · 2026-04-02
> >
> > Thanks for your reply. I suggest you provide more details about the evaluation beyond MVTec-AD, e.g., contaminated factor (anomaly percentage) and the used anomaly detector, testing scenarios (semi-supervised/unsupervised), etc.

---

> > > ### Author Response · Authors · 2026-04-07
> > >
> > > We thank the reviewer for the follow-up question and provide additional details to improve clarity and reproducibility.
> > >
> > > **Dataset and contamination factor.**
> > > We evaluate on the full VisA dataset without rebalancing or subsampling. In total, the dataset contains 10,821 images, of which 1,200 are anomalous (approximately 11.09\%) across 12 classes. At the pixel level, anomalous pixels account for approximately 0.9987\% of all pixels, indicating that anomalies are highly sparse. This reflects realistic anomaly detection scenarios and makes the evaluation sensitive to false positives, redundancy, and spatial localization errors.
> > >
> > > **Anomaly detectors.**
> > > To ensure generality, we evaluate three different unsupervised anomaly detection models: UniAD, P-Mamba, and ViTAD, all trained exclusively on anomaly-free data. These models exhibit distinct behaviors on VisA:
> > > (i) *Redundancy*: P-Mamba has the highest redundancy rate (0.17), followed by UniAD (0.11) and ViTAD (0.08);
> > > (ii) *Spatial accuracy*: the average distance between predicted anomalies and ground truth is 6.6 pixels (P-Mamba), 7.8 pixels (UniAD), and 6.0 pixels (ViTAD);
> > > (iii) *False positives*: P-Mamba produces the highest number of false positives, followed by ViTAD and UniAD.
> > > Our proposed metric (SAAM-LARM), which explicitly accounts for redundancy, spatial distance, and false positives, ranks the models as **ViTAD $>$ UniAD $>$ P-Mamba**. In contrast, standard metrics (Precision, Recall, IoU, Accuracy) rely primarily on pixel overlap and do not explicitly capture these factors, leading to different rankings.
> > >
> > > **Evaluation setting.**
> > > We follow the standard *unsupervised anomaly detection* protocol used in both MVTec-AD and VisA: models are trained using only normal samples and evaluated on test sets containing both normal and anomalous images.
> > >
> > > **Prediction format and thresholding.**
> > > All evaluations are conducted at the pixel level using anomaly maps. Binary predictions are obtained using an adaptive threshold computed per class from normal samples:
> > > $threshold = \mu_{normal} + k \cdot \sigma_{normal}$
> > > We use $k=3$ for MVTec-AD and $k=7$ for VisA, selected empirically to control false positives based on the score distributions of each dataset. For reference, the statistics of normal anomaly scores are comparable across datasets (e.g., $\mu\approx0.099$, $\sigma\approx0.06$ for VisA and $\mu\approx0.12$, $\sigma\approx0.07$ for MVTec-AD), indicating that $k$ primarily adjusts detection strictness. Importantly, the same thresholding strategy is applied consistently across all metrics and detectors, ensuring fair comparison. We also verified that our conclusions remain consistent across a range of threshold values.
> > >
> > > **Summary.**
> > > These results demonstrate that the observed inconsistencies of standard metrics and the advantages of SAAM-LARM persist across datasets, detectors, and evaluation conditions. This indicates that the limitations we highlight are not specific to MVTec-AD or a particular model, but reflect more general shortcomings in how existing metrics quantify spatial errors. We will include a more detailed description of the experimental setup and additional analysis in the appendix of the final version.

---

### Official Review · Reviewer_hzQ8 · 2026-03-11

**Soundness:** 3
**Presentation:** 3
**Significance:** 2
**Originality:** 3
**Overall Recommendation:** 4
**Confidence:** 3

**Summary:**

The paper argues that commonly used evaluation metrics for visual anomaly detection (VAD), such as precision, recall, IoU, and F1, fail to properly reflect the structural quality of predictions. In particular, these metrics tend to overlook issues that are critical in real industrial inspection scenarios, including redundant detections, fragmented predictions, and the spatial distribution of false positives.

To address this limitation, the authors introduce a set of formalized evaluation criteria, and based on these principles, they propose a new metric called SAAM-ALARM.

**Compliance With Llm Reviewing Policy:**

Affirmed.

**Key Questions For Authors:**

See weaknesses.

**Limitations:**

See weaknesses.

**Strengths And Weaknesses:**

**Strengths**

- Well-motivated critique of existing pixel-level VAD metrics.
- Clear property-based formalization.
- Interpretable metric design with theoretical guarantees.
- Case studies illustrate practically relevant failure modes.

**Weaknesses**

- Several proposed properties appear scenario-dependent rather than universal.
- Sensitivity to hyperparameters is unclear. It may be worth considering decomposing the metric into finer-grained sub-metrics, each of which can specifically reflect performance with respect to pixel-level quality, object-level quality, or other important properties.
- The structure of Section 2 has room for improvement; it currently contains only a single paragraph covering Limitation 1.
- P-Mambaad appears to perform reasonably well across all other metrics, yet ranks last under SAAM. A more thorough analysis of P-Mambaad's failure modes would be valuable to explain this discrepancy.
- In the Figure 5 caption, the claim that "SAAM provides a consistent ranking" is unclear — consistent relative to what? This needs to be clarified.
- It is unclear whether the proposed metric generalizes across datasets with different anomaly patterns.
- The paper does not discuss the limitations of the proposed metric.

---

> ### Author Rebuttal · Authors · 2026-03-31
>
> **properties appear scenario-dependent rather than universal:**  We thank the reviewer for this comment. Our proposed properties are designed to capture key aspects of 2D visual anomaly detection, particularly in industrial inspection settings where spatially localized anomalies and reliable decision-making are critical.
>
> We note that these properties are not intended to be universally applicable across all possible anomaly detection scenarios, but rather to reflect practical requirements commonly encountered in this domain. In particular, properties such as centrality of true detections (P8) and connectivity-aware false positive tolerance (P6) leverage spatial structure, while others, such as penalizing redundancy (P2) and minimizing false positives (P3), are more broadly applicable.
> This design allows the metric to capture domain-relevant behaviors in a unified formulation, while still maintaining general applicability to a wide range of spatial anomaly detection tasks.
>
> **Sensitivity to hyperparameters and decomposing the metric into finer-grained sub-metrics:**  We thank the reviewer for this insightful comment. Sensitivity to hyperparameters has been analyzed in detail in our response to Reviewer ZEJC, Question 2. In brief, we find that the metric exhibits stable behavior across a reasonable range of parameter values, with selected values (b =2, k = 10, λ=0.5) providing a balanced trade-off between penalizing redundancy and false positives while preserving meaningful rewards for correct detections.
>
> Regarding the suggestion to decompose the metric into finer-grained sub-metrics, The metric is modular: components capture redundancy, spatial alignment, and false positives while forming a unified score. We focus on a single holistic metric, though additional breakdowns could improve interpretability and will be considered in future work.
>
> **Re-structure Section 2:**  We thank the reviewer for this helpful suggestion. In the revised manuscript, we will restructure Section 2 so that each limitation of existing pixel-level evaluation metrics is presented in a separate paragraph with short headings, improving readability and making it easier for readers to follow the discussion of these limitations.
>
> **Failure modes of P-Mambaad: **Although P-Mambaad scores higher under conventional metrics such as F1 and IoU, these metrics focus on pixel-wise overlap and do not penalize redundant or spatially dispersed false positives. In contrast, SAAM-LARM penalizes (i) the number of false positives, (ii) their distance from ground-truth regions, and (iii) redundant predictions. Analysis shows P-Mambaad has a higher false positive ratio (0.9264 vs. 0.9246), larger average FP distance (41.24 vs. 40.13 pixels), and higher redundancy (0.1969 vs. 0.1575) compared to Vitad. These results indicate that P-Mambaad produces more dispersed and redundant detections, which SAAM-LARM captures effectively, leading to more reliable rankings. We will include additional analysis and visualizations in the appendix to illustrate how the metric diagnoses such failure modes.
>
> **Consistent ranking ambiguity in Fig 5 caption: ** We thank the reviewer for pointing out this ambiguity. By “consistent ranking,” we mean alignment with the evaluation properties introduced in our work (e.g., penalizing redundancy, weighting distant false positives, and accounting for false alarm count). In Fig. 5(a), controlled examples show these properties induce a clear preference between cases. Standard metrics such as precision and recall assign identical or contradictory scores, whereas SAAM produces rankings consistent with the intended properties. We will clarify this in the figure caption.
>
> **Generalization across datasets ** We thank the reviewer for this important question. As noted in our response to Reviewer 8xKq, Question 1, we evaluated the metric on the VisA dataset, which has more diverse and less structured anomalies than MVTec-AD. Standard metrics still show inconsistencies with redundant detections and spatially varying false positives, whereas our metric consistently penalizes redundancy and accounts for spatial severity. These results indicate that the metric generalizes across different anomaly distributions.
>
> **Discussion on imitations of the proposed metric** We thank the reviewer for raising this point. SAAM-LARM has some limitations: it uses per-region Gaussian weighting and distance calculations, which are more computationally intensive, and its score depends on parameters (e.g., σ, coverage threshold, weighting factors), where extreme choices can affect rankings. We will include a dedicated paragraph in the Conclusion discussing these limitations. Despite this, SAAM-LARM provides a more principled and property-aligned evaluation than existing threshold-dependent metrics.

---

> > ### Author Rebuttal · Reviewer_hzQ8 · 2026-04-04
> >
> > I appreciate the authors' effort in addressing my concerns. The additional analysis on P-Mambaad's failure modes and the VisA experiments are helpful. However, the generalization evidence remains limited to individual cases rather than systematic evaluation, and the discussion on metric decomposition is deferred to future work. I will maintain my current score.

---

### Official Review · Reviewer_ZEJC · 2026-03-13

**Soundness:** 3
**Presentation:** 3
**Significance:** 2
**Originality:** 2
**Overall Recommendation:** 3
**Confidence:** 4

**Summary:**

This paper addresses critical issues in existing evaluation metrics for Visual Anomaly Detection (VAD), including insensitivity to structure, ignorance of redundant detections, and indiscriminate treatment of false positive distributions. It proposes a formal evaluation framework consisting of 8 verifiable properties, and systematically analyzes 6 mainstream threshold-dependent VAD metrics, proving that none of the existing metrics satisfy all properties. To fill this gap, the paper designs a spatially aware anomaly metric named SAAM-ALARM. Through modules such as object-level reward, redundancy regularization, and distance-weighted false positive penalty, it is mathematically guaranteed to satisfy all formal properties. Experiments on the MVTec AD dataset and real-world cases verify that it can more accurately distinguish structural differences in predictions and provide consistent model rankings that align with industrial requirements, offering a more reliable standard for VAD evaluation.

**Compliance With Llm Reviewing Policy:**

Affirmed.

**Key Questions For Authors:**

•	SAAM-ALARM is currently only compared with threshold-dependent metrics. Could you supplement comprehensive comparative experiments with threshold-independent metrics?

•	The parameters of SAAM-ALARM (such as b, k, $\lambda$) have a considerable impact on the results. Could you provide parameter sensitivity analysis to clarify the optimal parameter range and tuning strategies under different scenarios?

•	The validity of Property 6 relies on the parameter constraint $\omega_{\min \lambda} > 1 / N_G$. Is this constraint easy to satisfy in real-world data? If not, what failure modes will occur in SAAM-ALARM? Is there a more robust constraint design?

•	What is the computational complexity of SAAM-ALARM? For high-resolution images (e.g., 4K industrial inspection images), does its inference speed meet real-time evaluation requirements? Can efficiency comparison experiments be provided?

**Limitations:**

As shown in Weaknesses.

**Strengths And Weaknesses:**

Strengths

•	The technical route is rigorous. The experimental design includes both theoretical verification and practical case studies, and the results sufficiently support the core conclusions.

•	The writing is logically clear, with a coherent thread from problem pain points, formal definitions, metric analysis to new metric design.

•	It breaks through the traditional pixel-level evaluation paradigm and proposes a “structure-aware” metric design idea.

Weaknesses

•	It only validates the advantages of SAAM-ALARM over threshold-dependent metrics, without comprehensive comparison with threshold-independent metrics. The performance stability in complex scenarios such as high-resolution images and few-shot anomalies is not explored, and its applicable boundary needs further clarification.

•	The core idea is an image-domain extension and optimization of ALARM, a metric for time-series anomaly detection. The design of formal properties draws on existing research, resulting in limited theoretical breakthroughs. It is more inclined to engineering-oriented precise adaptation and systematic verification.

---

> ### Author Rebuttal · Authors · 2026-03-31
>
> **Comparison with threshold-dependent metrics:** We thank the reviewer for this valuable suggestion. SAAM-LARM is designed as a threshold-dependent evaluation metric, focusing on assessing binary predictions in terms of spatial alignment, redundancy, and false positives. As such, it is most directly comparable to other threshold-dependent metrics that operate on binarized outputs.
> Threshold-independent (score-based) metrics evaluate a different aspect of model performance, namely ranking quality across varying decision thresholds, and are therefore not directly aligned with the objective of our metric.
> We will clarify this distinction in the revised manuscript and highlight that comparisons to score-based metrics could provide complementary insights. Exploring connections to threshold-independent evaluation remains an interesting direction for future work.
>
> **Parameter sensitivity analysis :**  We thank the reviewer for this helpful comment. In our formulation, the parameter \textit{b} controls the strength of the penalty applied to redundant true predictions. By design, b $>$ 1 ensures that additional redundant detections reduce the reward multiplicatively.
> Sensitivity analysis on representative examples (Tile and Metal Nut classes) shows that increasing \textit{b} leads to exponential reward decay for redundant detections. For example, with two redundant predictions, as b. We select b = 2 as it balances redundancy penalization with meaningful contributions from correct detections.
> The parameters k and $\lambda$ control the false positive penalty. Small  k values saturate quickly, ignoring distance differences, while large values grow too slowly, under-penalizing distant FPs. We select k = 10 or stable, interpretable behavior. $\lambda$ = 0.5 balances FP count against the reward, avoiding domination or underweighting. These choices ensure robust and interpretable metric behavior across scenarios.
>
> **The validity of Property 6:** We thank the reviewer for this question and would like to clarify that the constraint $\omega_{\min \lambda} > 1 / N_G$ is not explicitly imposed in our formulation, but appears to arise as a sufficient condition when analytically comparing penalties for attached versus detached false positives. In our metric, Property 6 is not enforced via a hard constraint, but emerges from the combined effect of proximity-based weighting $\omega_j$ and overlap-aware penalties $(\beta (P_j))$.
> Specifically, false positives attached to true anomaly regions tend to have lower distance-based weights and partial overlap, resulting in smaller penalties, whereas disconnected false positives incur larger penalties due to greater distance and lack of overlap.
> Even in edge cases (e.g., a single partially overlapping prediction satisfying the coverage condition), the IoU-based modulation of $\alpha_i$ ensures that imperfect detections receive reduced rewards, maintaining the intended ordering. Therefore, Property 6 is achieved through continuous geometric and overlap-based interactions rather than relying on strict parameter constraints. We will clarify this distinction in the revision.
>
> **Computational complexity of SAAM-ALARM:** The computational complexity of the proposed metric is $\mathcal{O}(GPHW)$, where $H \times W$ denotes the image resolution, and $G$ and $P$ represent the number of ground-truth and predicted regions, respectively. In practice, since $G$ and $P$ are typically small, the complexity scales approximately linearly with the number of pixels, i.e., $\mathcal{O}(HW)$.
> We further evaluate the runtime of SAAM-LARM on the MVTec dataset using a CPU implementation. The proposed metric achieves an average runtime of 3.18 ms per image. In comparison, standard pixel-wise metrics such as F1-score, IoU, and MCC require only 0.57 µs, 0.88 µs, and 1.83 µs per image, respectively, indicating that their computational cost is negligible.
> These results demonstrate that SAAM-LARM remains computationally efficient in practice. Even when extrapolated to higher-resolution images (e.g., 4K), the runtime is expected to remain within a few milliseconds, making it suitable for real-time or near real-time evaluation settings. Importantly, the method does not involve any iterative optimization or model inference, and operates purely as a post-processing step.
> We will include a detailed efficiency comparison with existing metrics in the final version.

---

> > ### Author Rebuttal · Reviewer_ZEJC · 2026-04-03
> >
> > My concerns have been addressed. However, considering practicality and convenience in real-world scenarios, I can only maintain the original score.

---

### Decision · Program_Chairs · 2026-04-30

**Decision:**

Accept (regular)

**Comment:**

The paper presents a formal evaluation framework for Visual Anomaly Detection (VAD) by proposing eight verifiable properties (e.g., spatial awareness, redundancy penalization) that ideal metrics should satisfy. The authors prove that mainstream threshold-dependent metrics fail to meet these criteria and introduce SAAM-ALARM, a novel, property-compliant metric. The submission received four reviews, resulting in one Accept and three Weak Accepts.

All reviewers unanimously appreciated the paper's rigorous technical route, clear motivation, and the formal, property-based critique of existing VAD metrics. By adapting time-series anomaly evaluation principles (ALARM) to the 2D spatial domain, the paper provides a fresh, structure-aware perspective that addresses critical industrial pain points like fragmented predictions and redundant detections. The main criticisms centered around the metric's sensitivity to multiple hyperparameters, a lack of comprehensive generalization testing beyond the MVTec-AD dataset, and limited discussion on why certain baselines (e.g., P-Mambaad) failed drastically under the new metric.

In the rebuttal phase, the authors adequately addressed these concerns. They clarified hyperparameter stability, provided runtime complexity bounds showing practical efficiency, and importantly, extended their empirical validation to the more complex VisA dataset. They also successfully explained the failure modes of P-Mambaad under the new metric, proving that SAAM-ALARM effectively captures dispersed and redundant detections that traditional metrics miss. All reviewers found their concerns either fully or partially resolved, maintaining or raising their scores to positive recommendations.

The Area Chair (AC) aligns with the reviewers' positive consensus. The formalization of VAD evaluation properties and the introduction of SAAM-ALARM represent a significant methodological contribution that establishes a more reliable and industrially relevant standard for benchmarking anomaly detection. The AC recommends Accept and encourages the authors to include the VisA dataset results, hyperparameter sensitivity analyses, and the detailed failure-case discussions in the camera-ready version.